# ACTG-ARL: Differentially Private Conditional Text Generation with RL-Boosted Control

## Abstract

Generating high-quality synthetic text under differential privacy (DP) is critical for training and evaluating language models without compromising user privacy. Prior work on synthesizing DP *datasets* often fail to preserve key statistical attributes, suffer utility loss from the noise required by DP, and lack fine-grained control over generation. To address these challenges, we make two contributions. First, we introduce a hierarchical framework that decomposes DP synthetic text generation into two subtasks: *feature learning* and *conditional text generation*. This design explicitly incorporates learned features into the generation process and simplifies the end-to-end synthesis task. Through systematic ablations, we identify the most effective configuration: a rich tabular schema as feature, a DP tabular synthesizer, and a DP fine-tuned conditional generator, which we term ACTG (**A**ttribute-**C**onditioned **T**ext **G**eneration). Second, we propose Anchored RL (ARL), a post-training method that improves the instruction-following ability of ACTG for conditional generation. ARL combines RL to boost control with an SFT anchor on best-of-$N$ data to prevent reward hacking. Together, these components form our end-to-end algorithm **ACTG-ARL**, which advances both the quality of DP synthetic text (+20% MAUVE over prior work) and the control of the conditional generator under strong privacy guarantees.

## 1 Introduction

Modern AI applications rely on vast amounts of user data, ranging from keyboard inputs on mobile devices (Hard et al., 2018; Xu et al., 2023; Zhang et al., 2025b) and recommender interaction histories (Karatzoglou and Hidasi, 2017; Zhang et al., 2019) to conversational preferences (Bai et al., 2022; Ouyang et al., 2022). This reliance poses significant privacy risks, which have become especially pressing with the rise of large language models (LLMs), as recent studies show they can memorize sensitive information from training corpora and expose it during user interactions (Carlini et al., 2021; Lukas et al., 2023; Nasr et al., 2025).

To maximize the value of data while preserving user privacy, a promising approach is to generate differentially private (DP) synthetic data (Hu et al., 2024). This paradigm offers a key advantage over task-specific DP mechanisms: it avoids the cumbersome need to design a new solution for each application. Instead, the DP synthetic dataset can be reused across any downstream task without incurring additional privacy cost or requiring changes to existing data pipelines. DP synthetic text, in particular, has attracted growing interest, spurring a line of work that harnesses the power of LLMs through fine-tuning or API-based prompting to continually push the frontier of the privacy-utility trade-off (Yue et al., 2023; Kurakin et al., 2023; Xie et al., 2024; Hou et al., 2024; Yu et al., 2024; Hou et al., 2025; Tan et al., 2025).

Despite these advances, most existing work on DP synthetic text remains limited to producing synthetic *datasets*, overlooking the critical need for *conditional generation* (Keskar et al., 2019; Dathathri et al., 2020). Conditional generation offers flexibility by enabling fine-grained control over the generation process, a capability of significant practical value. It allows users to synthesize data tailored to specific requirements (e.g., emails with positive sentiment) and enables analysts to preserve key statistical attributes or ensure balanced representation across subpopulations through controlled variations (Przystupa and Abdul-Mageed, 2019). Moreover, as we will see shortly, conditional

generation enables us to obtain not only private synthetic text but also private synthetic *features*, which can be valuable for a wide range of downstream analytical tasks.

In this work, we develop an integrated approach to DP (conditional) text generation that achieves high-quality synthetic text and fine-grained control under strong privacy guarantees. Specifically, our contributions are as follows:

- **A hierarchical framework for DP synthetic text generation.** We propose a framework that decomposes the problem of generating DP synthetic text into two subtasks: feature learning and conditional text generation. This modularity allows for systematic optimization, and through comprehensive ablations, we identify the most effective configuration: a rich tabular schema for feature, a specialized DP tabular synthesizer, and a DP fine-tuned conditional generator, which we collectively term **ACTG**[1] (**A**ttribute-**C**onditioned **T**ext **G**eneration).
- **Boosting fine-grained control in ACTG.** While ACTG produces high-quality synthetic datasets, we observe that its conditional generator suffers a significant loss in instruction-following ability under DP. To address this, we develop **Anchored RL** (ARL), a post-training recipe applied on top of ACTG. It combines a reinforcement learning (RL) objective to improve control with a supervised fine-tuning (SFT) objective on best-of-$N$ data, anchoring the model to the private text distribution and mitigating reward hacking.
- **State-of-the-art results in DP conditional text generation.** On challenging, real-world datasets, our integrated approach, **ACTG-ARL**, which combines ACTG with ARL, establishes a new state of the art. It *simultaneously* advances the quality of DP synthetic text over prior work (+20% in MAUVE and +50% in attribute distribution matching) while delivering a conditional generator with strong instruction-following capabilities. This enables fine-grained, controllable generation for diverse and practical applications.

## 2 PRELIMINARIES

We review the basics of differential privacy and DP synthetic data for both text and tabular domains.

**Differential Privacy (DP).** Our work builds on Differential Privacy (DP) (Dwork et al., 2006), the gold standard for privacy widely adopted in both government and industry (Erlingsson et al., 2014; Thakurta et al., 2017; Ding et al., 2017; Abowd, 2018). DP provides a mathematical guarantee that limits what an adversary can infer about any single user or record from an algorithm's output.

**Definition 2.1** (($\varepsilon, \delta$)-DP). A randomized algorithm $\mathcal{M}$ is ($\varepsilon, \delta$)-DP if for any two neighboring datasets $D$ and $D'$ which can be obtained from each other by adding or removing a single record, and for any subset of possible outputs $S$, the following inequality holds:
$$\mathbb{P}[\mathcal{M}(D) \in S] \leq e^\varepsilon \mathbb{P}[\mathcal{M}(D') \in S] + \delta.$$
Here, $\varepsilon$ is typically called the privacy budget where a smaller value implies stronger privacy protection, and $\delta$ represents the (small) failure probability.

**DP synthetic data.** DP synthetic data (Charest, 2011; Near and Darais, 2021) provides a privacy-preserving surrogate of the original dataset, aiming to retain core utility under formal DP guarantees. Owing to the post-processing property of DP (Dwork et al., 2014), such data can be freely shared and used for downstream tasks without additional privacy cost.

*Text data.* Since the seminal work of Yue et al. (2023), DP fine-tuning (DP-FT) has become the dominant approach for generating DP synthetic text (Kurakin et al., 2023; Yu et al., 2024; Tan et al., 2025). In this approach, a base model is fine-tuned on private text using a DP optimizer such as DP-Adam (Li et al., 2022). While Private Evolution (PE) (Lin et al., 2024; Xie et al., 2024) has emerged as a promising alternative, recent evidence indicates that DP-FT on a moderately sized model already outperforms PE (Tan et al., 2025). We defer more discussion of related work to Appendix B.

*Tabular data.* Generating high-dimensional synthetic tabular datasets under DP is a well-studied problem, with strong algorithms (e.g., AIM (McKenna et al., 2022)) and standardized benchmarks (Tao et al., 2021; Chen et al., 2025). A key design choice of our framework (Sec. 3.2) is to build on this foundation by recasting part of the synthetic text problem as a tabular synthesis task.

---

[1]The acronym draws on a biological metaphor: just as the four nucleotide bases (A, C, T, G) lie at the core of DNA, *feature* serves as the fundamental building block of our conditional generation framework.

## 3 A HIERARCHICAL FRAMEWORK FOR DP SYNTHETIC TEXT GENERATION

In this section, we introduce a modular, hierarchical framework for DP synthetic text generation. We detail its algorithmic design choices and present a comprehensive empirical evaluation to verify its effectiveness and ablate its core components.

### 3.1 A HIERARCHICAL FRAMEWORK

Our framework generalizes CTCL (Tan et al., 2025), which fine-tunes a conditional generator using *topics* as input, guided by a privatized topic histogram derived from a pretrained topic model. Compared to learning the private text distribution end-to-end, CTCL enjoys two advantages: i) histograms have low sensitivity and thus retain high utility under a tight privacy budget, and ii) conditioning simplifies the synthesis task, as topics guide the generator. These factors together yield a favorable privacy-utility trade-off and state-of-the-art results in *resource-constrained* DP synthetic text generation.

However, CTCL has several limitations that undermine its reliability in some settings. It relies on a fixed topic model trained on a public corpus, which may not align with the private domain and can force nuanced text into coarse, lossy categories, making the inferred topics inaccurate. Moreover, when the dataset size is small relative to the number of topics, the topic histogram contains many empty bins, yielding a low signal-to-noise ratio that renders the privatization step unstable (see Appendix F.4 for analysis). The combination of inaccurate topic inference and noise-dominated histograms can limit the utility of the generated text. This motivates us to generalize CTCL into a broader and more flexible hierarchical framework and systematically analyze its design choices.

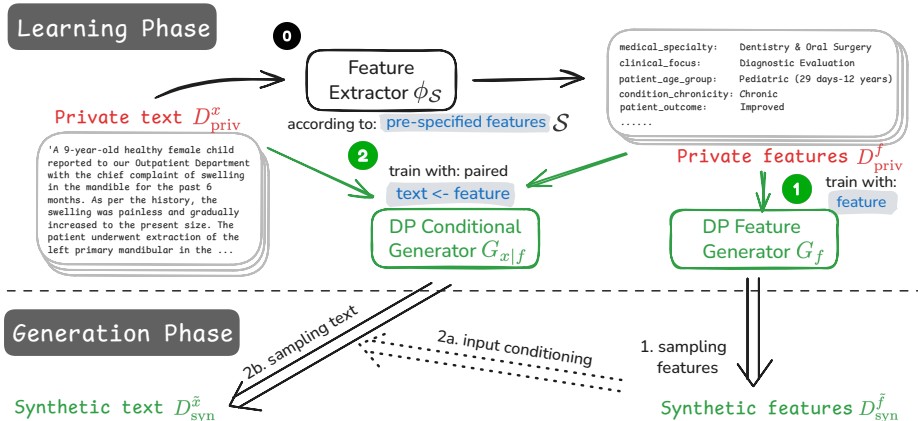

Figure 1: Our hierarchical framework for DP synthetic text generation.

We propose a modular, hierarchical framework that *decomposes* DP synthetic text generation into two subtasks: learning a *low-dimensional* feature representation of private text, and learning a conditional generator. Given a feature design $\mathcal{S}$, Fig. 1 shows how the **Learning Phase** unfolds in three stages:

- *Stage 0: Feature extraction.* We employ a feature extractor $\phi_{\mathcal{S}} : x \to f$ to extract the private feature set $D_{\text{priv}}^f$ from the raw text corpus $D_{\text{priv}}^x$ according to $\mathcal{S}$, where $x$ denotes a text sample and $f$ its feature. We also compose the (feature, text) pairs into $D_{\text{priv}}^{f,x}$. This pre-processing step prepares the data for subsequent stages.
- *Stage 1: Learning a DP feature generator.* We learn a generator $G_f$ with privacy budget $\varepsilon_1$ on $D_{\text{priv}}^f$. The goal is to generate synthetic features that resemble the private ones.
- *Stage 2: Learning a DP conditional generator.* We learn a conditional generator $G_{x|f}$ with privacy budget $\varepsilon_2$ on $D_{\text{priv}}^{f,x}$. The goal is to generate synthetic text that resemble the private text, while adhering to the requirements specified by the features.

In the **Generation Phase**, once $G_f$ and $G_{x|f}$ are obtained, DP synthetic text can be produced without further access to private data. We first sample synthetic features from the DP feature

generator $\tilde{f} \sim G_f$, and then feed them into the DP conditional generator to produce the final output $\tilde{x} \sim G_{x|f}(\cdot \mid \tilde{f})$. We refer to the synthetic DP feature set as $D_{\text{syn}}^{\tilde{f}}$ and the synthetic DP text set as $D_{\text{syn}}^{\tilde{x}}$.

**Remark 3.1.** *Stage 0 (feature extraction) is a pre-processing step and does not consume any privacy budget (we treat the feature extractor as a trusted component and defer more discussions to Appendix C.1). The overall privacy guarantee of our framework is $(\varepsilon, \delta)$, obtained by composing the budgets of Stage 1 ($\varepsilon_1$) and Stage 2 ($\varepsilon_2$). We rely on advanced composition for privacy accounting and defer detailed descriptions to Sec. 3.3.1 and Appendix C.2.*

**Remark 3.2.** *CTCL (Tan et al., 2025) is an instantiation of our framework: the feature $\mathcal{S}$ is topic, the feature extractor $\phi_{\mathcal{S}}$ is a topic model, the DP feature generator is a privatized topic histogram, and the DP conditional generator is a DP fine-tuned language model pretrained on a public dataset.*

## 3.2 Instantiation of the Framework

While CTCL represents one concrete instantiation, our framework's modularity defines a rich design space with a wide range of algorithmic choices. We exploit this flexibility to conduct comprehensive ablation studies and identify an optimal configuration. In what follows, we explore this design space across three key dimensions:

**Feature design and extraction.** We explore three distinct feature designs ($\mathcal{S}$), each requiring a specialized feature extractor ($\phi_{\mathcal{S}}$):

- ($\mathcal{S}_1$) **Topic** (CTCL, Tan et al. (2025)): A single topic defined by a list of keywords, extracted using the pretrained topic model from their work ($\phi_{\mathcal{S}_1}$)[2].

- ($\mathcal{S}_2$) **Free-form summary:** A concise summary of the text with 1-2 sentences, generated by a powerful LLM $M_{\text{oracle}}$, which serves as the extractor ($\phi_{\mathcal{S}_2}$).

- ($\mathcal{S}_3$) **Structured tabular schema:** A rich, multi-attribute schema with fixed options per attribute, treated as tabular data and annotated by $M_{\text{oracle}}$ ($\phi_{\mathcal{S}_3}$). Unlike a set of fixed, generic topics, the schema is *dataset-specific* and designed to capture the key dimensions of the data. We describe the LLM-assisted process for schema design and feature extraction in Appendix C.1

**DP feature generator** ($G_f$). The choice of the feature generator is related to the feature design. For $\mathcal{S}_1$, CTCL used a privatized histogram. For $\mathcal{S}_2$ and $\mathcal{S}_3$, we consider two types of feature generator: DP-FT (Yue et al., 2023) and AIM (McKenna et al., 2022). DP-FT is broadly applicable to any feature that can be represented in a textual format, encompassing both free-form and schema-based features. In contrast, AIM is designed specifically for tabular data, where features must be either categorical with finite options or numerical values.

**DP conditional generator** ($G_{x|f}$). For conditional text generation, we consider two approaches: 1) performing DP-FT on a base LLM on the paired (feature, text) set $D_{\text{priv}}^{f,x}$, such that it learns to generate synthetic text conditioned on the input feature, or 2) prompting a powerful LLM ($M_{\text{gen}}$), leveraging its strong instruction-following capabilities.

## 3.3 Experiments

We conduct a comprehensive empirical evaluation to demonstrate the effectiveness of our framework against strong baselines and to assess the impact of its core components through detailed ablations. We first outline our experimental setup (Sec. 3.3.1) and then present and discuss the results (Sec. 3.3.2).

### 3.3.1 Experimental setup

**Datasets.** We conduct experiments on two challenging, domain-specific datasets: **bioRxiv** (Hou et al., 2025), a corpus of scientific abstracts ($n = 29$k, average number of tokens per sample around 300), and **PMC-patients** (Zhao et al., 2023), a collection of sensitive clinical notes ($n = 240$k, average tokens per sample around 450). These represent a more difficult testbed than

---

[2]Their pretrained topic model can be downloaded from https://github.com/tanyuqian/synthetic-private-data

the general-domain corpora (e.g., Yelp (Yelp, Inc., 2025)) frequently used in prior work (Yue et al., 2023; Xie et al., 2024; Tan et al., 2025). We provide further details for both datasets in Appendix E.1.

**Baselines.** We compare against three baselines: Aug-PE (Xie et al., 2024), vanilla DP-FT (Yue et al., 2023), and CTCL (Tan et al., 2025). While DP-FT has been reported as a weak baseline when implemented with GPT-2 (Radford et al., 2019), we find its performance improves substantially with stronger base models, consistent with observations in Kurakin et al. (2023); Yu et al. (2024). CTCL (corresponding to $\mathcal{S}_1$) was originally studied under resource-constrained settings, and we adapt it to our setup with a larger model, though without additional pretraining. Finally, we refer to the two proposed designs, $\mathcal{S}_2$ and $\mathcal{S}_3$, as *our conditional generation approaches*.

**Implementation details.** For fair comparison, we use the same base model `gemma-3-1b` `-pt`[3] (Team et al., 2025) for all methods that require fine-tuning (vanilla DP-FT, CTCL, and our approaches), and a more powerful instruction-tuned model `Qwen2.5-7B-Instruct`[4] (Qwen et al., 2024) for Aug-PE. We use `gemini-2.5-flash-lite`[5] both as the oracle model for feature extraction ($M_{\text{oracle}}$) and as the conditional generator for prompting ($M_{\text{gen}}$). Further discussion of model choices and additional implementation details are provided in Appendix E.2.

**Privacy budget and accounting.** We evaluate all methods under three total privacy budgets: $\varepsilon \in \{1, 4, \infty\}$. Following standard practice (Yue et al., 2023; Xie et al., 2024), we set $\delta = 1/(n \log n)$, where $n$ is the size of the private training set. The total privacy cost of our framework is the composition of the budgets for the feature generator ($\varepsilon_1$) and the conditional generator ($\varepsilon_2$). For each method and each total budget $\varepsilon$, we independently tune the budget split ($\varepsilon_1, \varepsilon_2$). Full details on our privacy accounting are in Appendix C.2.

**Evaluation suite.** Our evaluation suite assesses data quality along multiple dimensions, capturing both broad and specific aspects of synthetic text. First, for general *fidelity*, we follow prior work (Yue et al., 2023; Xie et al., 2024) and use MAUVE (Pillutla et al., 2021) to quantify semantic similarity between the synthetic and private text distributions. For *utility*, we measure the F1 score on a downstream classification task, and the next token prediction (NTP) accuracy on a downstream generation task. Finally, we evaluate fine-grained *attribute distribution matching* according to the features in $\mathcal{S}_3$. Specifically, we define $d_{\text{JS}}^f$ as the Jensen-Shannon distance between the private and synthetic feature distributions (extracted from $D_{\text{priv}}^x$ and $D_{\text{syn}}^{\tilde{x}}$), averaged over all attributes considered. This metric captures discrepancies in feature distributions that matter for downstream analysis, reflecting what an analyst might care about in practice. In Appendix F.5, we further evaluate topic distribution matching, which goes beyond schema attributes to assess the alignment in broad topic structure. Detailed descriptions and implementations of all metrics are provided in Appendix E.3.

**Remark 3.3.** *We highlight several limitations in existing evaluation practices. First, prior studies (Yue et al., 2023; Xie et al., 2024; Yu et al., 2024; Tan et al., 2025) often rely on relatively weak embedding models (e.g., all-MiniLM-L6-v2 or sentence-T5) when computing MAUVE, which can inflate the scores and obscure quality differences. Second, they adopt a short context length (e.g., 128 or 256 tokens), discarding content beyond the cutoff and failing to assess the overall quality of text. In our evaluation, we address both issues by employing stronger, domain-specific embedding models for MAUVE and adopting a longer context window. A more detailed discussion is provided in Appendix F.1.*

### 3.3.2 EXPERIMENTAL RESULTS

**Comparison with baselines.** We begin by comparing the end-to-end performance of our conditional generation approaches with the baselines. For $\mathcal{S}_2$, we use DP-FT for both the feature and conditional generator, while for $\mathcal{S}_3$ we adopt the optimal configuration ACTG, which we will discuss shortly in the ablation studies. As shown in Fig. 2 (rows 1 and 2), our methods consistently outperform Aug-PE, vanilla DP-FT and CTCL across all datasets, privacy levels, and evaluation metrics. These results provide strong empirical support for the core hypothesis of our framework: *decoupling feature learning and conditional text generation yields a superior privacy-utility trade-off.*

**Comparison of feature design $\mathcal{S}$.** We next compare the three feature designs within our framework. Results show that the rich tabular schema ($\mathcal{S}_3$) performs best, followed by the free-form summary ($\mathcal{S}_2$),

---

[3] https://huggingface.co/google/gemma-3-1b-pt
[4] https://huggingface.co/Qwen/Qwen2.5-7B-Instruct
[5] https://ai.google.dev/gemini-api/docs/models#gemini-2.5-flash-lite

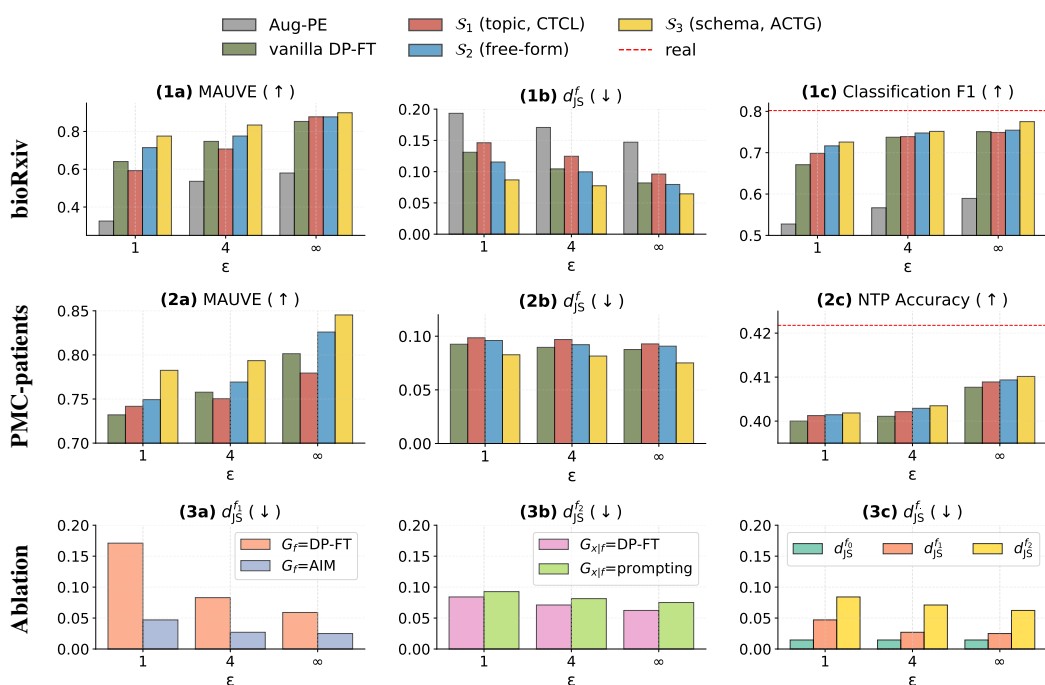

Figure 2: **End-to-end and modular evaluation of our hierarchical framework. (Rows 1–2)**
End-to-end comparison of our approaches with baselines (Aug-PE, vanilla DP-FT, CTCL) on bioRxiv
and PMC-Patients, evaluated on fidelity (MAUVE, $d_{JS}^f$) and utility (classification F1, NTP accuracy).
We omit Aug-PE on PMC-Patients in Row 2 (see full results in Appendix F.3) as its performance
is substantially lower than the other methods. **(Row 3)** Modular ablations and fine-grained error
analysis for $S_3$. Arrows in the figure titles indicate whether higher ($\uparrow$) or lower ($\downarrow$) values are better.

both substantially outperforming the topic model ($S_1$) used in CTCL. Unlike $S_3$ and $S_2$, which are
tailored to each private dataset, the generic topic model in CTCL can suffer from domain mismatch;
this underscores the importance of domain-specific features. Moreover, the superiority of the tabular
schema $S_3$ over free-form text $S_2$ highlights the value of a compact yet informative schema that
captures key information about the private dataset with minimal bits. This observation resonates with
the notion of *compact representation* discussed in Hu et al. (2024).

**Ablation studies.** We focus on $S_3$ and conduct a modular evaluation to dissect errors at different
stages. Fig. 3 illustrates three sources of error: **extraction error** ($d_{JS}^{f_0}$) from LLM-based annotations
of private text; **feature learning error** ($d_{JS}^{f_1}$) introduced in Stage 1; and **conditional generation error**
($d_{JS}^{f_2}$) introduced in Stage 2. Measuring extraction error primarily serves to validate the reliability
of LLM annotations, which form the basis of our evaluation, while analyzing feature learning and
conditional generation errors enables us to identify the optimal configurations for these stages.

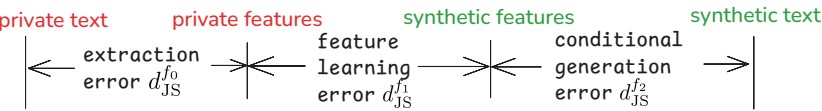

Figure 3: Fine-grained error analysis in our framework.

*Evaluation approach.* For $d_{JS}^{f_0}$, since ground-truth features of $D_{priv}^x$ are unavailable, we perform five
independent extractions and treat their *average distribution* as ground truth. We then compute the
Jensen-Shannon distance between each trial and this average, and average across all attributes. For
$d_{JS}^{f_1}$, we compute the distance between $D_{priv}^f$ and $D_{syn}^{\tilde{f}}$. Finally, for $d_{JS}^{f_2}$, we compute the distance
between $D_{priv}^{\tilde{f}}$ and the attributes extracted from $D_{syn}^{\tilde{x}}$.

*Results.* We present ablation results on bioRxiv in Fig. 2. In **Stage 0**, the extraction error $d_{\text{JS}}^{f_0}$ is around 0.01, confirming that LLM-extracted features are reliable. For **Stage 1**, we compare AIM and DP-FT as feature generators. Fig. 2(3a) shows that AIM achieves a much lower $d_{\text{JS}}^{f_1}$. One key reason is that AIM, as a specialized tabular synthesizer, allocates privacy budget only to predefined attributes of interest, rather than across all tokens as in DP-FT. This avoids wasting budget on non-sensitive information (e.g., JSON grammar) or public knowledge (e.g. age groups), thus improving the privacy-utility trade-off. Finally, Fig. 2(3b) shows that in **Stage 2**, DP-FT achieves a lower $d_{\text{JS}}^{f_2}$ than direct prompting. We discuss the drawbacks of direct prompting in detail in Appendix F.2.

Taken together, these ablations empirically confirm our *optimal* configuration: a rich structured tabular schema ($\mathcal{S}_3$), AIM feature generator ($G_f$), and a DP-FT conditional generator ($G_{x|f}$). We henceforth refer to this configuration as **ACTG: Attribute-Conditioned Text Generation**, which establishes a new state of the art in DP synthetic text generation.

We also provide a comparative error analysis across the three stages of ACTG. As shown in Fig. 2(3c), extraction error is negligible, while conditional generation incurs a larger error than feature learning, suggesting greater room for improvement in Stage 2. We will revisit this point in Sec. 4.3.

# 4 BOOSTING FINE-GRAINED CONTROL IN ACTG WITH ANCHORED RL

So far, we have developed a general framework and identified its optimal configuration, ACTG, which produces high-quality DP synthetic *datasets*. However, generating static datasets is only part of the story. In practice, users may also require *controlled, on-demand generation*—producing text that satisfies specific requirements, such as an email with a positive tone on a given topic. In this setting, the focus shifts from *aggregate* dataset-level metrics like fidelity and utility to the generator's *per-instance* ability to reliably follow instructions.

In this section, we study the instruction-following capability of ACTG's conditional generator $G_{x|f}$ and show that it is significantly degraded under DP. To address this challenge, we propose Anchored RL, a post-training method built upon ACTG that strengthens control while preserving alignment with $D_{\text{priv}}^x$. Importantly, using RL to enhance control is made possible by the conditional generation framework in Sec. 3 and ACTG's design, where tabular features serve as explicit, verifiable rewards.

## 4.1 MEASURING AND IMPROVING INSTRUCTION FOLLOWING

We focus on the structured tabular schema ($\mathcal{S}_3$, with $K$ fields) used in ACTG and introduce the metric of **instruction following accuracy** (IFAcc). For a given input $f$, the generator $G_{x|f}$ produces a text $x$, from which a feature $\hat{f}$ is extracted using $M_{\text{oracle}}$. The *per-instance* IFAcc is defined as the fraction of fields in $\hat{f}$ that correctly match the input instruction $f$, and the overall IFAcc is the average of these per-instance scores across all text features. Formally:

$$\text{IFAcc} := \mathbb{E}_{f \sim D_{\text{priv}}^f} \left[ \frac{1}{K} \sum_{k=1}^{K} \mathbb{I}(f_k = \hat{f}_k) \right], \tag{1}$$

where $\mathbb{I}(\cdot)$ is the indicator function and the expectation is taken over the private feature set.

Evaluating the conditional generator $G_{x|f}$ in ACTG, we find its instruction-following accuracy is significantly degraded by DP (e.g., dropping from 66% to 53% on bioRxiv; see Fig. 4(a)). This loss of fine-grained control, even when aggregate metrics remain high, motivates a post-training procedure to restore instruction-following in conditional generation.

**Boosting control via RL.** Because ACTG is built on tabular features, it provides a natural interface for reinforcement learning. Each input feature $f$ serves as a *rubric*[6] for scoring generations. For each generated text $x$, we compute the per-instance IFAcc as the reward. The training loop is straightforward: we sample prompts from the DP feature generator ($f \sim G_f$), generate text ($x \sim G_{x|f}$), and use the resulting reward to update the model. We term this approach built on top of (and enabled by) ACTG as **ACTG-RL**. Crucially, unlike Wu et al. (2024a) who privatize the policy

---

[6]This *rubric-as-reward* paradigm has recently gained significant interest (Gunjal et al., 2025; Viswanathan et al., 2025; Huang et al., 2025).

**(a)** DP degrades IFAcc **(b)** RL hurts MAUVE   **(c)** Example of reward hacking: TL;DR-style generation

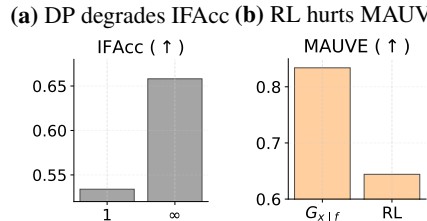

**Input feature:** {"primary_research_area": "Neuroscience", "model_organism": "Drosophila melanogaster", "experimental_approach": "Wet Lab Experimentation", "dominant_data_type": "Phenotypic / Behavioral", "research_focus_scale": "Cellular", "disease_mention": "No Specific Disease Mentioned", "sample_size": "Relies on Cell/Animal Replicates", "research_goal": "Investigating a mechanism"}

**Generated abstract:** We experimentally evaluated whether larval synaptic plasticity is preserved by modulating spatial memory formation in *Drosophila*.

Figure 4: **(a)** IFAcc of the conditional generator $G_{x|f}$ with and without DP, showing a substantial drop under DP. **(b)** MAUVE score of generated text after RL, demonstrating a sharp decline in textual fidelity. **(c)** Example generation from the bioRxiv dataset that perfectly satisfies the input requirement (score: 8/8; see Appendix F.12) but fails to match the target domain (paper abstract). This occurs during RL training, where the model exploits the rubric reward and exhibits reward hacking.

gradients, our RL training phase requires no additional privacy budget as both the prompts and the reward signal are derived without accessing the private data.

**The reward hacking issue.** ACTG-RL adopts the standard PPO objective (Schulman et al., 2017), which reveals a clear trade-off between control and fidelity. While instruction following accuracy (IFAcc) improves, the MAUVE score plummets (see Fig. 4(b)), indicating a significant loss of textual quality. A closer examination of the outputs reveals the failure mode: the model learns to *hack* the reward by generating short "TL;DR"-style sentences. These outputs perfectly satisfy the rubric's criteria but fail to resemble the target domain's style (e.g., a full abstract); we present an example of such a generation in Fig. 4(c). This failure highlights the need for a method that can boost control without sacrificing textual alignment. We address the challenge in what follows.

### 4.2 A POST-TRAINING RECIPE: ANCHORED RL

We introduce Anchored RL (ARL), a post-training recipe for boosting instruction-following without sacrificing alignment with the original text distribution. Our approach makes two key design choices: a *hybrid training objective* to balance control and textual alignment, and a method for curating a high-quality *synthetic anchor dataset* without additional privacy cost.

**Hybrid objective.** Inspired by standard practices in RLHF (Bai et al., 2022; Ouyang et al., 2022), our core idea is to mitigate reward hacking by anchoring the model to a reference distribution using a supervised fine-tuning (SFT) loss. The training objective thus becomes a hybrid of the standard RL loss and this SFT loss. However, this raises a critical question: what data should be used for the SFT anchor? Using the original private data would incur additional privacy cost, while the synthetic data sampled from $G_{x|f}$ suffers from the control issues that we aim to fix.

**High-quality anchor via best-of-$N$ sampling.** We resolve this by crafting a high-quality, private dataset for the SFT objective using *best-of-$N$ sampling*, a technique that has been widely used in aligning LLMs (Gao et al., 2023; Eisenstein et al., 2024). For each feature $f$ sampled from the DP generator $G_f$, we generate $N$ candidate texts from $G_{x|f}$, and select the one with the highest per-instance IFAcc score. This process leverages additional test-time compute to distill a much cleaner dataset, $D_{\text{SFT}}$, which provides a strong signal for the SFT anchor without incurring additional privacy cost. We provide a detailed analysis of the quality of the best-of-$N$ dataset and the variance of per-instance IFAcc scores in Appendix F.13.

**Putting together: Anchored RL.** Our final training recipe, ARL, combines these two components. We fine-tune from the DP-FT checkpoint $G_{x|f}$ using a hybrid objective that mixes the PPO gradient with the SFT gradient on curated best-of-$N$ data: $\mathcal{L} = \mathcal{L}_{\text{RL}} + \gamma \cdot \mathcal{L}_{\text{SFT}}$. We employ a *linear decay* schedule for the coefficient $\gamma$, starting high to preserve text fidelity and gradually decreasing to allow for steady improvement in instruction following. This approach anchors the model, preventing it from drifting away from the desired text distribution while optimizing for control.

**The end-to-end algorithm.** Our final algorithm, **ACTG-ARL**, integrates ACTG with ARL into a single cohesive pipeline. It consists of four stages: private feature extraction, training the initial DP generators, curating the anchor dataset, and performing ARL training. The outputs are a DP synthetic

dataset and a DP conditional generator with strong instruction-following capabilities. A detailed description is provided in Alg. 1 of Appendix D.

## 4.3 EXPERIMENTS

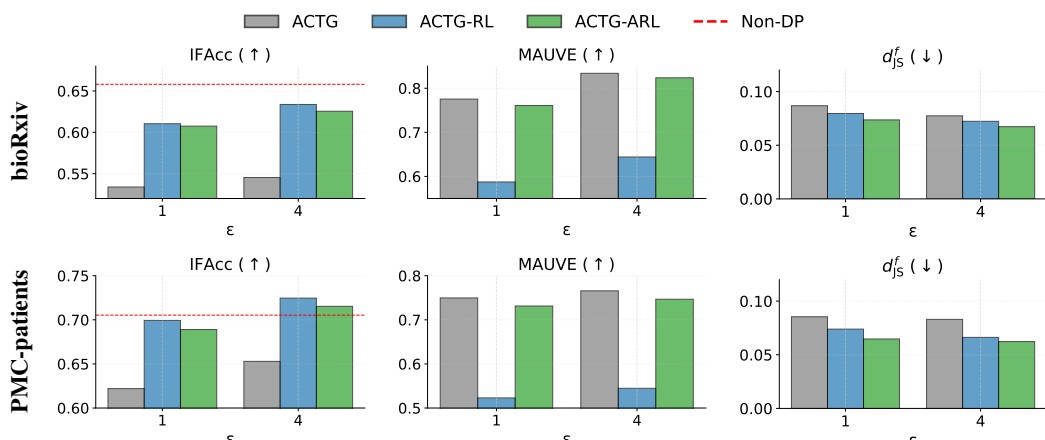

Figure 5: **Performance of the conditional generators before and after RL evaluated on three metrics.** ACTG-RL improves IFAcc but suffers from reward hacking, which collapses textual fidelity (MAUVE). ACTG-ARL resolves this trade-off, boosting IFAcc close to the non-DP level while maintaining high MAUVE and achieving the best attribute distribution matching.

We evaluate three models: 1) ACTG, the conditional generator $G_{x|f}$ from Sec. 3 which serves as the baseline; 2) ACTG-RL, where $G_{x|f}$ is further trained with a standard PPO objective; and 3) ACTG-ARL, where $G_{x|f}$ is trained with the hybrid objective and the best-of-$N$ anchor dataset $D_{\text{SFT}_N}$, corresponding to our Anchored RL recipe. In Appendix F.6, we also consider a variant of ACTG where the base PT model is replaced with an IT model for DP-FT. In Appendix F.14, we showcase the importance of $D_{\text{SFT}_N}$ and RL via ablation studies. Details of experimental setups are in Appendix E.4.

**Results.** Fig. 5 illustrates the central trade-off between control and fidelity. The baseline ACTG achieves strong fidelity (high MAUVE) but weak control (low IFAcc). ACTG-RL increases IFAcc but suffers from reward hacking, causing a catastrophic collapse in MAUVE. In contrast, ACTG-ARL resolves this tension: it matches ACTG-RL in IFAcc while retaining the high fidelity of ACTG. Importantly, this improvement in *per-instance* control also reduces the *end-to-end* error, as evidenced by the lowest $d_{\text{JS}}^f$. Since $d_{\text{JS}}^f$ is a *metric* (Lin, 2002), the triangle inequality implies $d_{\text{JS}}^f \leq d_{\text{JS}}^{f_1} + d_{\text{JS}}^{f_2}$, shedding light on why advances in Stage 2 reduce overall error, echoing our discussion in Sec. 3.3.

## 5 CONCLUSION AND LIMITATIONS

We introduced a hierarchical framework (with ACTG as the optimal configuration) and a novel Anchored RL recipe that, together, form our end-to-end algorithm **ACTG-ARL**. This approach yields: 1) state-of-the-art DP synthetic text datasets, and 2) a controllable, instruction-following generator. Beyond these concrete results, our work advances the field by elevating *control* as a third, crucial dimension alongside utility and privacy in DP synthetic text generation—one that brings broad practical benefits.

A limitation of our study is that all experiments were conducted on a fixed model size (`gemma-3-1b-pt`). An important future work is to explore how our algorithm performs across model scales: scaling up to test whether the gains persist, and scaling down to understand the smallest model size that can benefit from design choices such as rich schema, best-of-$N$ sampling, and RL. Another promising direction is to explore how pre-generation conditioning, which is the focus of this work, can be combined with post-generation filtering or resampling as in Yu et al. (2024), to further improve the quality of synthetic datasets.

ETHICS STATEMENT

This work advances synthetic data generation under formal differential privacy guarantees, and we do not foresee additional ethical or privacy risks.

REPRODUCIBILITY STATEMENT

Our code and datasets are publicly available at https://github.com/actg-arl/ACTG-ARL.

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

## A  THE USE OF LARGE LANGUAGE MODELS (LLMS)

We used large language models (LLMs) solely for minor text editing and polishing, such as improving clarity, grammar, and flow. All conceptual development, technical content, experiments, and analyses were conducted entirely by the authors.

## B  RELATED WORK

**DP synthetic text.**  Research on DP synthetic text generation is largely divided into two paradigms: DP fine-tuning (DP-FT) (Bommasani et al., 2019; Mattern et al., 2022; Yue et al., 2023; Carranza et al., 2024; Ochs and Habernal, 2025) and Private Evolution (PE) (Xie et al., 2024; Hou et al., 2024; 2025). Fine-tuning-based approaches learn the private text distribution implicitly via next-token prediction, while PE-based approaches leverage the power of LLMs to create a large pool of samples and iteratively refine them using distance to the private data in the embedding space. A recent trend in fine-tuning-based approaches is to improve data quality through *distribution alignment*. This can be done *post-hoc*, for instance by filtering synthetic data (Yu et al., 2024), or *a priori*, by using topics to condition the generation process (Tan et al., 2025). Our work builds upon and significantly extends this conditional approach: we abstract the concept into a general hierarchical framework and enhance it with a novel post-training RL recipe to improve fine-grained control.

**Conditional text generation.**  Existing approaches to conditional text generation often rely on low-level signals to steer a model's output. For example, Putta et al. (2023) adjust a model's hidden states using feedback from an attribute classifier, while DeSalvo et al. (2024) employ soft prompts to guide generation. Although CTCL (Tan et al., 2025) uses conditioning, its reliance on a fixed, general-purpose topic model presents two limitations for specialized datasets. First, the general model might perform poorly on these datasets due to distribution mismatch. Second, many of the predefined topics might be irrelevant and consequently receive zero counts in the topic histogram. In contrast, our framework uses human-interpretable, domain-specific features, offering more flexible and transparent control over the generated text.

## C  ADDITIONAL DETAILS OF OUR APPROACH

### C.1  LLM-ASSISTED SCHEMA DESIGN AND EXTRACTION

**Schema design via LLM.**  Fig. 6 provides the initial prompt for schema design. Using the template prompt, we fill in the corresponding "dataset_description", "workload_description", and "num_features" for each private dataset domain we want to build a schema for. We optionally supply examples if public or donated examples are available.

**Discussion on feature extraction.**  For $\mathcal{S}_1$, we rely on a pretrained topic model for feature extraction. Since this model can be downloaded and run locally, it poses no risk of privacy leakage. For $\mathcal{S}_2$ and $\mathcal{S}_3$, we instead use a powerful LLM, $M_{\text{oracle}}$ (specifically `gemini-2.5-flash-lite`), to perform feature extraction.

In our main experiments, we assume a threat model in which the server-hosted model is trustworthy. This means that sharing data with the server does not lead to a privacy breach, consistent with the policies of major LLM providers[7]. Nevertheless, even if the server behaves adversarially (Duan et al., 2023b), our algorithm remains applicable. In such cases, one option is to deploy an open-source LLM locally for the same task. Alternatively, privacy-preserving inference methods can be adopted to enable secure API queries (Duan et al., 2023a; Wu et al., 2024b; Tang et al., 2024; Hong et al., 2024).

**Schemas for the two datasets.**  We derive schemas for the two private datasets in our study following the above and provide their corresponding schema below: Fig. 7 for bioRxiv and Fig. 8 for PMC-patients.

---

[7]https://ai.google.dev/gemini-api/terms#data-use-paid

I would like to extract structured features from unstructured data. Your task is to analyze the dataset description and provided examples to define a set of representative categorical features.
# Dataset Description
{data_description}
# Primary Goal
The extracted features should be optimized to be as useful as possible for the following workload:
{workload_description}
# Core Task
Generate a set of {num_features} categorical features that provide a rich summary of the underlying text.
# Feature Requirements:

1. **Feature Diversity**: The feature set should be comprehensive. Strive to include a mix of:

    - **General-Purpose Features**: Attributes applicable to almost any text (e.g., Formality, Sentiment).
    - **Domain-Specific Features**: Attributes that capture the unique jargon, entities, or processes of the target dataset, keeping the data shift in mind.

2. **Orthogonality**: Prioritize features that are orthogonal / independent, unless they are intentionally hierarchical.

3. **Values**: Each feature must have a fixed set of at most 50 explicitly enumerated possible values. These values must be representative of the target data. Use an "Other" category where appropriate.

4. **Hierarchical Features**: Conditional features are permitted. If a feature's relevance depends on the value of another, its value should be "Not Applicable" when the condition is not met (e.g., a 'LegalSubTopic' feature is only applicable if 'MainTopic' is 'Legal').

5. **Avoid Triviality**: Do not create features that are overly simplistic or too specific to a single exemplar.

# Output Format:
Provide your response as a numbered list. For each feature, you MUST include its name, possible values, a description, and a rationale for its inclusion.

1. **Feature Name**:

    • **Possible Values**: ...
    • **Description**: A brief, clear explanation of what the feature captures.
    • **Rationale**: A justification for why this feature is useful for the primary goal, citing an example if helpful.

2. ...

# Examples:
{_formatted_examples}

Figure 6: A detailed prompt for schema identification. We fill in {data_description}, {workload_description}, {num_features} for each dataset based on general knowledge of the dataset domain. For {_formatted_examples}, this field is optional and we supply a few examples publicly available in the general domain.

**Feature extraction via LLM.**     After obtaining the schema, we prompt $M_{\text{oracle}}$ to extract the features according to the pre-specified schema. Fig. 9 provides the prompt for feature extraction.

```
{
    "primary_research_area": "< Biochemistry | Bioinformatics | Biophysics | Cancer
Biology | Cell Biology | Clinical Trials | Developmental Biology | Ecology | Epidemiology |
Evolutionary Biology | Genetics | Genomics | Immunology | Microbiology | Molecular Biology |
Neuroscience | Paleontology | Pathology | Pharmacology and Toxicology | Physiology | Plant
Biology | Public Health | Scientific Communication and Education |Structural Biology | Synthetic
Biology | Systems Biology | Zoology | Other>", // Categorizes the abstract into its main biological
discipline.
    "model_organism": "< Human | Mouse/Rat | Zebrafish | Drosophila melanogaster |
Caenorhabditis elegans | Saccharomyces cerevisiae | Escherichia coli | Arabidopsis thaliana
| Plant | Cell Culture | In Silico / Computational | Other Mammal | Other Vertebrate | Other
Invertebrate | Other Microbe | Not Applicable / Review | Other >", // Identifies the primary biological
model used in the research.
    "experimental_approach": "< Wet Lab Experimentation | Computational / In Silico
Analysis | Clinical Study | Field Study / Observation | Case Study / Case Review | Review /
Meta-analysis | New Method Development | Theoretical Modeling | Other >", // Describes the
main methodology used to conduct the study.
    "dominant_data_type": "< Genomic | Transcriptomic | Proteomic | Metabolomic |
Imaging | Structural | Phenotypic / Behavioral | Ecological / Environmental | Clinical / Patient
Data | Simulation / Model Output | Multi-omics | Other >", // Specifies the primary type of data
generated or analyzed in the paper.
    "research_focus_scale":        "<Molecular|Cellular|Circuit  /  Network|Tissue  /
Organ|Organismal|Population|Ecosystem|Multi-scale|Other>", // Categorizes the biological level of
organization the study focuses on.
    "disease_mention": "< Cancer | Neurodegenerative Disease | Infectious Disease |
Metabolic Disease | Cardiovascular Disease | Autoimmune / Inflammatory Disease | Psychiatric /
Neurological Disorder | Genetic Disorder | No Specific Disease Mentioned | Other >", // Identifies
whether the abstract explicitly names a disease or a major disease category.
    "sample_size": "< Single Subject / Case Study | Small Cohort (<50 subjects) | Medium
Cohort (50-1000 subjects) | Large Cohort / Population-scale (>1000 subjects) | Relies on
Cell/Animal Replicates | Not Specified / Not Applicable >", // Estimates the scale of the study based
on mentions of sample or cohort size.
    "research_goal": "< Investigating a mechanism | Characterizing a system/molecule |
Developing a method/tool | Identifying novel elements | Testing a hypothesis | Quantifying a
parameter | Evaluating/Comparing approaches | Other >" // Categorizes the study's primary objective
based on its framing.
}
```

Figure 7: Schema for the bioRxiv dataset

## C.2 PRIVACY ACCOUNTING

For each method and each total budget $\varepsilon$, we independently tune the budget split $(\varepsilon_1, \varepsilon_2)$. We use state-of-the-art privacy accountants (Mironov, 2017; Gopi et al., 2021; Doroshenko et al., 2022) to calibrate to the final $(\varepsilon, \delta)$-DP guarantee. We use different methods for accounting depending on the framework instantiation.

- **CTCL** ($\mathcal{S}_1$): composition of 1) a Gaussian mechanism (for the DP histogram) with 2) composition of subsampled Gaussian mechanism (for DP-FT). This can be handled with Privacy Loss Distribution (PLD) accountants (Gopi et al., 2021; Doroshenko et al., 2022).

- **Conditional generation with free-form feature** ($\mathcal{S}_2$): composition of 1) composition of subsampled Gaussian mechanism (for DP-FT) and 2) composition of subsampled Gaussian mechanism (for DP-FT). This can be handled with Privacy Loss Distribution (PLD) accountants (Gopi et al., 2021; Doroshenko et al., 2022).

```
{
    "medical_specialty": "< Cardiology | Dermatology | Dentistry & Oral Surgery |
Endocrinology | Gastroenterology | Hematology | Infectious Disease | Nephrology | Neurology |
Obstetrics & Gynecology | Oncology | Ophthalmology | Orthopedics | Otolaryngology (ENT)
| Pediatrics | Psychiatry | Pulmonology | Rheumatology | Surgery | Urology | Other >", // The
primary medical discipline. Map sub-specialties (e.g., Neurosurgery) to their primary field.
    "clinical_focus": "< Diagnostic Evaluation | Therapeutic Intervention | Monitoring
& Follow-up | Adverse Event >", // The narrative's primary purpose (e.g., diagnosis, intervention,
monitoring).
    "patient_age_group": "< Neonate (0-28 days) | Pediatric (29 days-12 years) | Adoles-
cent (13-17 years) | Adult (18-64 years) | Geriatric (65+ years) >", // The patient's specific age
category.
    "condition_chronicity": "< Acute | Chronic | Acute-on-Chronic | Recurrent / Re-
lapsing | Congenital >", // The temporal nature and pattern of the patient's primary condition.
    "narrative_structure": "< Chronological History | Problem-Oriented Summary |
Procedural Report | Other >", // The organizational style and flow of the clinical summary.
    "primary_intervention_type": "< Pharmacological | Surgical / Procedural | Sup-
portive & Conservative Care | Not Applicable >", // The primary therapeutic or management action
described (excluding diagnostic tests).
    "diagnostic_certainty": "< Confirmed Diagnosis | Provisional Diagnosis | Differen-
tial Diagnosis | Not Applicable>", // The level of diagnostic confidence expressed within the narrative.
    "patient_outcome": "< Resolved | Improved | Stable / Unchanged | Deteriorated |
Deceased | Referred / Transferred | In-Progress / Unknown >" // The patient's clinical status or
disposition at the end of the report.
}
```

Figure 8: Schema for the PMC-patients dataset

- **ACTG** ($\mathcal{S}_3$): AIM satisfies $\rho$-zCDP. From the perspective of the RDP accountant, this is interchangeable with a Gaussian machanism (as they have the same RDP guarantees for all $\alpha$). Thus we treat this as a composition of 1) a Gaussian mechanism (for AIM) and 2) composition of subsampled Gaussian mechanism (for DP-FT), and use RDP accountant (Mironov, 2017; Wang et al., 2019) to perform accounting.

### C.3 DETAILS OF THE RL ALGORITHM PPO

**Proximal Policy Optimization (PPO).** PPO (Schulman et al., 2017) is a policy gradient algorithm that maximizes a clipped surrogate objective to stabilize policy updates. Let $\pi_\theta(a \mid s)$ be the policy parameterized by $\theta$, and let $A^\pi(s, a)$ be an estimator of the advantage function. Define the probability ratio

$$r_\theta(s, a) = \frac{\pi_\theta(a \mid s)}{\pi_{\theta_{\text{old}}}(a \mid s)}.$$

The PPO objective is

$$\mathcal{L}^{\text{PPO}}(\theta) = \mathbb{E}_{(s,a) \sim \mathcal{B}} \Big[ \min\big(r_\theta(s, a) \, A^\pi(s, a), \ \text{clip}(r_\theta(s, a), \, 1 - \epsilon, \, 1 + \epsilon) \, A^\pi(s, a)\big) \Big],$$

where $\mathcal{B}$ is a batch of transitions collected using the old policy $\pi_{\theta_{\text{old}}}$, and $\epsilon > 0$ controls the amount of clipping to limit policy changes and promote stable learning.

**PPO in language models.** In LLM training, PPO is commonly implemented through the TRL (Transformer Reinforcement Learning) framework (von Werra et al.), which adapts the standard PPO update to sequence models by operating on token-level log-probabilities and performing rollouts in text space. In our implementation, we follow the TRL PPO pipeline but *replace the default LM-based reward model with our rubric reward* as detailed in Sec. 4.1. We further adapt the pipeline to optimize a hybrid objective of our Anchored RL (Sec. 4.2). We provide implementation details and experimental setups of ACTG-RL and ACTG-ARL in Appendix E.4.

> You are an expert biomedical information extraction assistant. Your task is to carefully read a scientific abstract from bioRxiv and extract the specified features according to the schema provided.
>
> Output exactly one JSON object with no extra text or explanations.
>
> **CRITICAL INSTRUCTION 1:** For any field in the JSON schema that lists specific options (e.g., "<Option1|Option2|...>"), you MUST select one of the provided options exactly as it is written. Do not invent, alter, or combine options. Failure to use an exact option from the list will be considered an error.
>
> **CRITICAL INSTRUCTION 2:** Ensure the value chosen for a field is appropriate for that field's specific definition. Do not use an option from one field (e.g., 'Cellular' from 'research_focus_scale') as the value for another field.
>
> Use this schema:
> ```json
> {schema}
> ```
>
> **Abstract to analyze:**
>
> {abstract_text}
>
> **Your output (JSON only):**

Figure 9: Prompt for feature extraction on the bioRxiv dataset, where `{schema}` is substituted with the content in Fig. 7 and `{abstract_text}` is the private text to be annotated.

## D    FULL ALGORITHM AND PSEUDOCODE

We provide the pseudocode of the full algorithm in Alg. 1, which includes the following stages.

1. **Private data annotation**: First, we annotate the private dataset with a structured tabular schema ($\mathcal{S}_3$) via inference calls to $M_{\text{oracle}}$.

2. **Initial DP generators training:** We then train the initial DP generators: the feature generator ($G_f$) using AIM, and the conditional text generator ($G_{x|f}$) using DP-FT.

3. **Anchor dataset curation:** Using the initial generators, we curate a high-quality synthetic dataset $D_{\text{SFT}_N}$ via best-of-$N$ sampling.

4. **Anchored RL:** We fine-tune the initial generator $G_{x|f}$ using Anchored RL, which combines an RL objective on prompts from $G_f$ with an SFT objective on the anchor dataset $D_{\text{SFT}_N}$. This leads to the final model $G_{x|f}^{\text{ARL}}$.

The procedure yields two key outputs: 1) a DP synthetic dataset, produced by sampling from $G_f$ and $G_{x|f}^{\text{RL}}$, and 2) a conditional generator $G_{x|f}^{\text{ARL}}$ with good instruction-following capabilities, which can be used for on-demand, targeted generation tasks.

## E    ADDITIONAL EXPERIMENTAL SETUPS

### E.1    DATASETS

We adopt two challenging, real-world datasets for our studies.

---

**Algorithm 1:** ACTG-ARL

---

**Input** : Private dataset $D_{\text{priv}}^x$, LLM feature extractor $M_{\text{oracle}}$, `AIM` parameter $\rho$, `DP-FT` parameter $\sigma$, best-of-N parameter $N$

**Output**: Feature generator $G_f$, final conditional text generator $G_{x|f}^{\text{ARL}}$, generated synthetic features $D_{\text{syn}}^{\tilde{f}}$, generated synthetic text $D_{\text{syn}}^{\tilde{x}}$

// Step 1:  Annotate private data

1 $D_{\text{priv}}^f \leftarrow \text{Annotate}(M_{\text{oracle}}, D_{\text{priv}}^x)$       ▷ Extract features according to a rich schema

// Step 2:  Train initial DP feature generator and conditional generator

2 $G_f \leftarrow \text{AIM}(D_{\text{priv}}^f, \rho)$       ▷ Obtain a DP feature generator for synthetic features

3 $G_{x|f} \leftarrow \text{DP-FT}(D_{\text{priv}}^x, D_{\text{priv}}^f, \sigma)$       ▷ Obtain an initial conditional text generator

// Step 3:  Generate the best-of-N SFT dataset, and the feature dataset for RL

4 $D_{\text{SFT}_N} \leftarrow \text{Best-of-N-Sampling}(G_f, G_{x|f}, N, M_{\text{oracle}})$       ▷ Perform best-of-N sampling

5 $D_{\text{RL}}^{\tilde{f}} \leftarrow \text{Sampling}(G_f)$       ▷ Sample features as input to RL

// Step 4:  Perform Anchored Reinforcement Learning

6 $G_{x|f}^{\text{ARL}} \leftarrow \text{AnchoredRL}(G_{x|f}, D_{\text{RL}}^{\tilde{f}}, D_{\text{SFT}_N})$       ▷ Train from $G_{x|f}$ using a weighted sum of losses

// Final step:  Generate synthetic text

7 $D_{\text{syn}}^{\tilde{f}} \leftarrow \text{Sampling}(G_f)$       ▷ Sample synthetic features

8 $D_{\text{syn}}^{\tilde{x}} \leftarrow \text{Sampling}(G_{x|f}^{\text{ARL}}, D_{\text{syn}}^{\tilde{f}})$       ▷ Sample synthetic text conditioned on synthetic features

9 **return** $G_f, G_{x|f}^{\text{ARL}}, D_{\text{syn}}^{\tilde{f}}, D_{\text{syn}}^{\tilde{x}}$

---

**bioRxiv** (Hou et al., 2025) is a dataset of abstracts on the bioRxiv preprint server. The raw dataset is hosted on HuggingFace[8]. We filter the dataset to contain only the abstracts appearing after the knowledge cutoff date of Gemma-3 family models (Aug 2024)[9]. We perform train/validation/test split and obtain a train set of size $n = 28,846$. We examine the token length for samples in the dataset, and found that the $95\%$ quantile is $512$ tokens. Thus we use context length $512$ tokens with all methods.

**PMC-patients** (Zhao et al., 2023) is a large-scale dataset of clinical notes documenting the patients' clinical visits. By nature, this dataset is highly sensitive. Upon examination, we found that only basic anonymity techniques were applied on the released dataset (e.g. redacting the patient's name). We use the latest dataset offered on HugggingFace[10] (version V2, released in 2024). We perform train/validation/test split and obtain a train set of size $n = 240,294$. We again use context length of $512$ tokens.

### E.2 IMPLEMENTATIONS DETAILS FOR THE HIERARCHICAL FRAMEWORK AND BASELINES

**Aug-PE.** We set the number of PE iterations $T$ to 10 and number of variations $L$ to 7 for both datasets, and perform privacy accounting using the code provided by the authors[11]. Following the recommendation in their paper (Xie et al., 2024), we perform selection by rank after the histogram voting.

**DP-FT.** We use the same code base of DP-FT, for all methods involving DP-FT as its subcomponents (vanilla DP-FT, as well as our conditional generation approaches ). The codebase is adapted from Yu et al. (2024)[12]. Below we describe the hyperparameters: we use batch size $b = 2048$, iterations $T = 1120$ for bioRxiv (80 epochs) and $T = 1170$ for PMC-patients (10 epochs). We set clipping

---

[8] https://huggingface.co/datasets/hazylavender/biorxiv-abstract
[9] https://ai.google.dev/gemma/docs/core/model_card_3
[10] https://huggingface.co/datasets/zhengyun21/PMC-Patients
[11] https://github.com/AI-secure/aug-pe/blob/main/notebook/dp_budget.ipynb
[12] https://github.com/google-research/google-research/tree/master/dp_instructions/dp_finetuning

norm $c$ to 1. We tune the learning rate $\eta \in \{$1e-4, 3e-4, 1e-3$\}$ and learning rate scheduler $\in\{$constant, cosine$\}$. We perform LoRA fine-tuning (Hu et al., 2022) with a LoRA rank of 8, $\alpha = 16$ and dropout of 0.05. The same set of hyperparameters are used for vanilla DP-FT as well as DP-FT within our framework (for training $G_f$ or $G_{x|f}$).

**CTCL (as $\mathcal{S}_1$ in our framework).** CTCL operates in the resource-constrained regime. We specifically adapt it to our setup: 1) Switch from an $O(100M)$ encoder-decoder model to `gemma-3-1b-pt`, i.e., the same base model as used in our approaches. 2) Drop the pretraining stage in the original paper for the encoder-decoder model. The authors pretrained a topic model and released the model checkpoint on their GitHub[13]; we directly used the checkpoint for topic feature extraction. We follow their paper (Tan et al., 2025) to set the noise multiplier in the Gaussian mechanism (for the DP histogram) as $\sigma = 10$, and use $H = 0$ for thresholding the noisy histogram.

**Sampling from a trained LLM.** We perform nucleus decoding (Holtzman et al., 2020) to sample from trained LLMs and adopt the following hyper-paramters: temperature $T = 1.0$, top-$p = 0.95$, top-$k = 0$. We sample $N_{\text{syn}} = 5,000$ for all methods except for Aug-PE where we use $N_{\text{syn}} = 2,000$ following Xie et al. (2024)[14].

### E.3 A COMPREHENSIVE EVALUATION SUITE

**MAUVE.** We follow Xie et al. (2024); Tan et al. (2025) and use MAUVE to measure textual alignment. We use the code provided by Pillutla et al. (2021)[15].

- *Embedding model:* We identify several issues with the usage of embedding models in the current literature and provide detailed discussions in Appendix F.1. We choose specialized sentence embedding models that are suitable for each dataset. For bioRxiv, we choose `SPECTER2` (Singh et al., 2022)[16] which is finetuned on abstracts of scientific papers. For PMC-patients, we adopt `S-PubMedBert-MS-MARCO` (Gu et al., 2020)[17] which is finetuned on full-text PubMed articles. Both embedding models have context length (`max_seq_length`) of 512, same as the context length of the models we fine-tune,

- *Number of clusters*: We follow the recommendation of the authors[18] to use 1/10 of the synthetic data size as the number of clusters, i.e., 500 for evaluating all other methods, and 200 for evaluating Aug-PE generated samples. We additionally comment that when evaluating the same dataset, using a smaller number of clusters will lead to a higher MAUVE score; this is because the resulting cluster is coarser.

**Classification F1.** We craft a classification task to measure the utility of the synthetic data in the following way. We introduce a new attribute, and then use $M_{\text{oracle}}$ to annotate both the real private test set and the generated data. After preparing the classification labels, we train on the synthetic data, and test on real data; this follows the standard evaluation approach considered in Yue et al. (2023); Xie et al. (2024); Tan et al. (2025).

- *Attribute and labels*: For bioRxiv, the new attribute is "research domain" with 8 meta-categories: {Biochemistry & Molecular Biology, Cell & Developmental Biology, Physiology & Immunology, Neuroscience & Cognition, Microbiology, Ecology & Evolution, Applied & Medical Biology, Computational Biology & Bioinformatics}.

- *Model*: We finetune a SciBERT model (Beltagy et al., 2019)[19] for the classification task. We perform full fine-tuning.

---

[13]https://github.com/tanyuqian/synthetic-private-data

[14]This is due to the high cost of sampling in Aug-PE: $T$ iterations where $N_{\text{syn}} \cdot (L - 1)$ samples are generated in each iteration, plus $N_{\text{syn}} \cdot L$ samples generated in the first iteration.

[15]https://github.com/krishnap25/mauve

[16]https://huggingface.co/allenai/specter2

[17]https://huggingface.co/pritamdeka/S-PubMedBert-MS-MARCO

[18]https://krishnap25.github.io/mauve/

[19]https://huggingface.co/allenai/scibert_scivocab_uncased

- *Metric*: We use macro-F1 (unweighted average across classes), due to the class imbalance naturally present in the dataset (see Fig. 10).

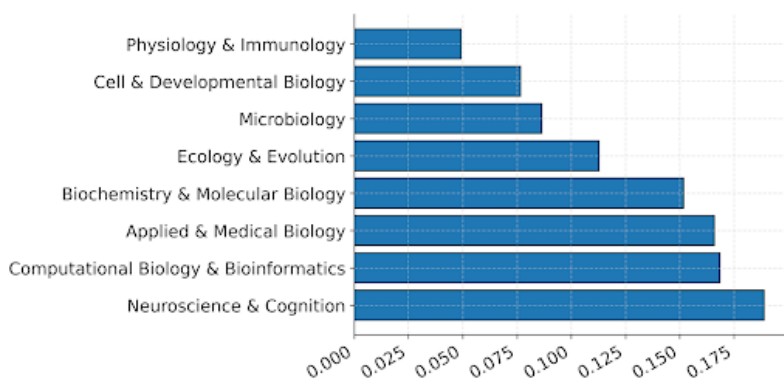

Figure 10: Histogram of labels of the "research domain" attribute in bioRxiv.

**Next token prediction (NTP) accuracy.** We fine-tune a small LM on the synthetic data, and test on real private data. We follow the literature (Xie et al., 2024; Tan et al., 2025) to fine-tune BERT-small and use the same set of standard hyperparameters.

**Attribute distribution matching.** We compute **Jensen-Shannon distance** for each attribute separately and average over the attributes. The obtained score represents attribute distribution matching.

- *Jensen-Shannon divergence (JSD)*. Given two probability distributions $P$ and $Q$ defined over the same domain, the Jensen-Shannon *divergence* (JSD) is defined as
$$\text{JSD}(P \parallel Q) = \tfrac{1}{2}\,\text{KL}(P \parallel M) + \tfrac{1}{2}\,\text{KL}(Q \parallel M), \quad M = \tfrac{1}{2}(P+Q),$$
where $\text{KL}(P \parallel Q) = \sum_x P(x) \log \frac{P(x)}{Q(x)}$ denotes the Kullback-Leibler divergence. JSD is a smoothed version of KL that can handle potential support mismatch.

- *Jensen-Shannon distance* is given by the square root of the Jensen-Shannon divergence:
$$d_{\text{JS}}(P,Q) := \sqrt{\text{JSD}(P \parallel Q)}.$$
This metric is symmetric, bounded in $[0,1]$, and widely used to quantify similarity between probability distributions.

## E.4 IMPLEMENTATION DETAILS FOR ACTG-RL AND ACTG-ARL

**RL specific setups.** We adopt TRL[20] as the RL fine-tuning framework and use the PPO objective (supported in `PPOTrainer`). We adapt the codebase from Singhal et al. (2024)[21] which provides an interface supporting different types of reward signals. For our ACTG-RL and ACTG-ARL, we implement the reward signal as the score from the LLM $M_{\text{oracle}}$ grading on the rubric, and integrate it into the RL fine-tuning pipeline.

**Training hyperparameters.** We use a rollout buffer size of $512$, batch size of $512$ and ppo epochs of $4$ (meaning we loop over the same rollout buffer for $4$ times). We set the learning rate $\eta = 5 \times 10^{-6}$ and train for $1000$ rounds in all. We set the initial KL coefficient to $0.2$. For ACTG-ARL specifically, we introduce $\gamma$, a mixing coefficient for the RL and SFT objective. We adopt a linear decay schedule for $\gamma$, staring from a bigger $\gamma$ for stabilizing the anchor and gradually decreasing it to allow for steady improvement in instruction following. For bioRxiv, we use a start value of 2 and ending value of 0.5. For PMC-patients, we use a start value of 0.5 and ending value of 0.2.

---

[20]https://huggingface.co/docs/trl/en/index
[21]https://github.com/PrasannS/rlhf-length-biases

## F  ADDITIONAL EXPERIMENTAL RESULTS

### F.1  ISSUES WITH MAUVE EVALUATION IN THE LITERATURE

We highlight several critical issues with MAUVE evaluation in the literature (Xie et al., 2024; Tan et al., 2025), which has gone unnoticed.

**Limited context length.**   We report the default context length of common sequence embedding models (adopted in Xie et al. (2024)) in Table 1. Note that all of the models have rather short context length ($< 512$). Because text beyond `max_seq_length` is truncated, these models can only evaluate quality with respect to a short prefix of the synthetic text. As a result, conclusions drawn from such biased evaluations may be significantly undermined.

| Embedding models in Xie et al. (2024) | Default context length max_seq_length |
|---|---|
| `sentence-t5-xl` | 256 |
| `sentence-t5-base` | 256 |
| `stsb-roberta-base-v2` | 75 |
| `all-MiniLM-L6-v2` | 256 |
| `paraphrase-MiniLM-L6-v2` | 128 |

Table 1: Default maximum sequence length for common sequence embedding models.

**Capability of the embedding model.**   We compute the MAUVE score of the same synthetic dataset w.r.t. embeddings extracted by different sequence embedding models. As shown in Fig. 11, weaker embedding models (`stsb-roberta-base-v2`, `all-MiniLM-L6-v2`) can inflate the MAUVE score, while a stronger embedding model (`Qwen3-Embedding-4B` (Zhang et al., 2025a)) is more capable of assessing the quality of the synthetic data.

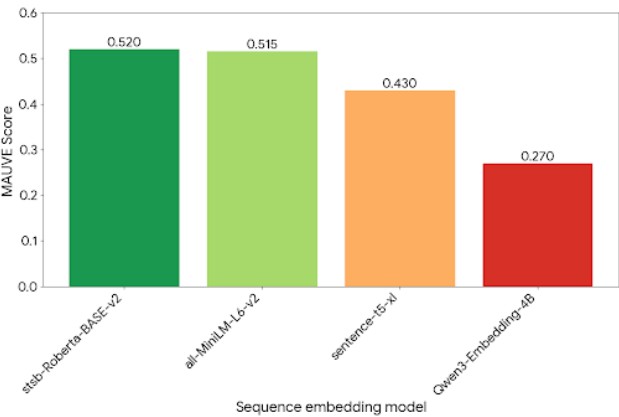

Figure 11: MAUVE score of the same synthetic dataset evaluated by different sequence embedding models.

### F.2  LIMITATIONS OF DIRECT PROMPTING AS THE CONDITIONAL GENERATOR

**Low MAUVE score.**   Fig. 12 shows that the MAUVE score of the direct prompting approach is extremely low. This is understandable as this approach does not have any access to the private text information, thus the poor textual alignment.

**Inherent bias and distribution mismatch.**   Fig. 13-(left) shows that the oracle tends to overemphasize certain categories (e.g., Cell Biology), generating disproportionately more samples in these

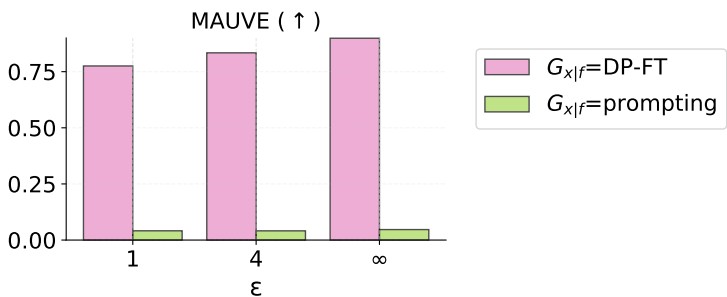

Figure 12: MAUVE scores achieved by different conditional generation approaches.

bins. Additionally, Fig. 13-(right) illustrates its inability to handle ambiguous or underspecified cases, leading to systematic errors when the input falls into categories such as "Other" or "Not Specified". These limitations stem from the fact that direct prompting has no means to calibrate feature distributions, as it handles each input feature independently. In contrast, our conditional generator obtained via DP-FT explicitly learns the mapping, enabling better attribute distribution matching.

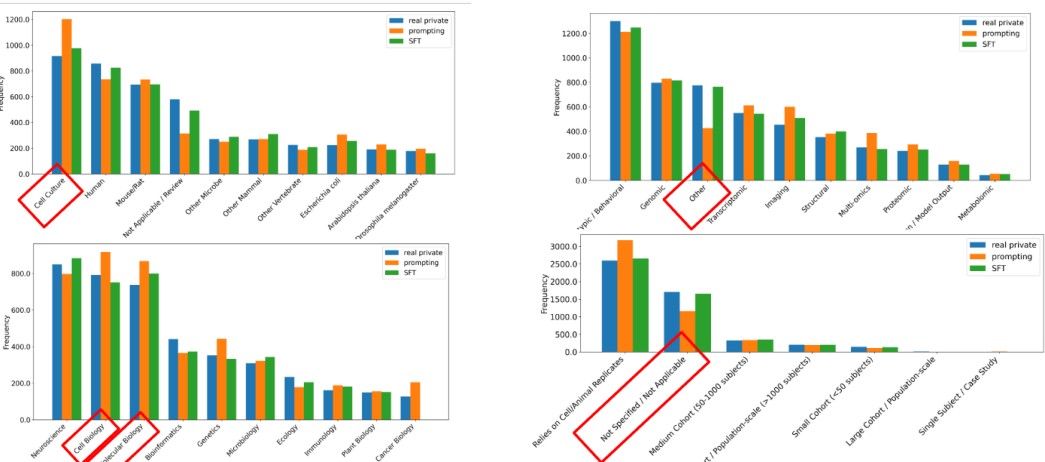

Figure 13: **(Left)** $M_{\text{oracle}}$ favors concept related to Cell Biology and generates disproportionally more samples categorized to it. **(Right)** $M_{\text{oracle}}$ fails to appropriately handle input of "Other" / "Not Specified".

### F.3 AUG-PE ON PMC-PATIENTS

Aug-PE fails to produce meaningful results on PMC-patients. Even in the non-private setting, it achieves a MAUVE score below $0.05$, attribute distribution matching $d_{\text{JS}}^f$ higher than 0.2, and NTP accuracy of 0.32. Across all metrics, its performance lags far behind other methods. These negative results are likely caused by the large distribution shift between the PMC-patients dataset and the public corpus of pretraining.

### F.4 FAILURE MODE OF CTCL

**Domain mismatch in topic extraction.** CTCL relies on a pretrained topic model trained on general-domain corpora (specifically, Wikipedia). Such models can perform poorly when applied to narrow, domain-specific datasets, such as clinical notes (e.g., PMC-patients in our study). As illustrated in Fig. 14, a dental case is associated with unrelated keywords like "fossil" and "paleontology". Although these associations may arise from shared biological or anatomical terminology, they clearly fail to capture the intended clinical meaning. This mismatch underscores the sensitivity of topic-based extraction to distribution shift and its limited effectiveness in specialized domains.

> **Example Text:** 'A 9-year-old healthy female child reported to our Outpatient Department with the chief complaint of swelling in the mandible for the past 6 months. As per the history, the swelling was painless and gradually increased to the present size. The patient underwent extraction of the left primary mandibular in the first molar for the same reason without any associated trauma, pain, or fever. Medical history was nonsignificant, with no history of systemic illness or long-term medication. On extraoral examination, there were no signs or symptoms. Intraorally, a diffuse swelling extending buccally from the distal of the primary mandibular left primary canine to the distal of the primary mandibular second primary molar ... (further text omitted)
>
> **Keywords associated with the predicted topic:** 'fossil, paleontology, dinosaur, fossils, jurassic, phylogeny, cretaceous, phylogenetic, dinosaurs, prehistoric'

Figure 14: Example of spurious topic associations in CTCL. A clinical note for a dental visit is linked to keywords such as "fossil".

**Sparse DP histogram.** When the number of samples is small relative to the number of topic bins (as in our bioRxiv dataset, where the dataset size is $N = 28{,}846$ but the number of topics is $1{,}827$), many bins can be empty. As all bins receive DP noise, when using a clipping threshold of zero (as in Tan et al. (2025)), the noise can dominate. Consequently, as shown in Fig. 15, the DP histogram deviates significantly from the real distribution. This distortion harms both the fidelity and utility of synthetic text as reflected in Fig. 2 in Sec. 3.

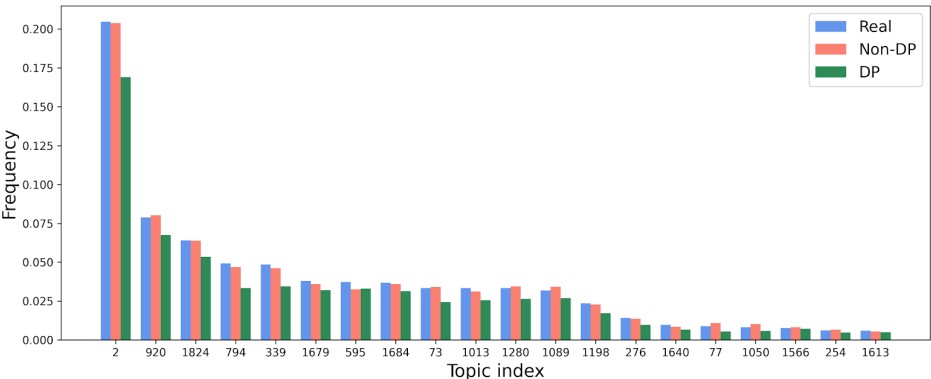

Figure 15: Comparison of topic histograms (top 10 topics, obtained on bioRxiv) of real data, non-DP samples, and DP samples. The non-DP histogram closely follows the real distribution, while the DP histogram diverges significantly due to noise amplification on sparse bins.

### F.5 TOPIC DISTRIBUTION MATCHING

We further evaluate whether synthetic data preserves the topic distribution of the private dataset. We train a topic model using FASTopic (Wu et al., 2024c)[22] with $n_{\text{topics}} = 50$ on the *private training set*, ensuring that the model captures domain-specific topic structure. The trained model is then applied to both the private test set and the synthetic datasets to obtain their corresponding topic distributions. We then compute the Jensen-Shannon distance ($d_{\text{JS}}^{\text{topic}}$) between the two distributions.[23] As shown in Fig. 16, the benefits of attribute conditioning carry over to topic distribution alignment, with ACTG showing the closest match to the private data.

---

[22] https://github.com/bobxwu/FASTopic

[23] Unlike the pre-trained topic model in Tan et al. (2025) which was trained on general-domain corpora (Wikipedia), the FASTopic model here is trained directly on the private training set, providing the most faithful characterization of the private data, which is critical for this evaluation.

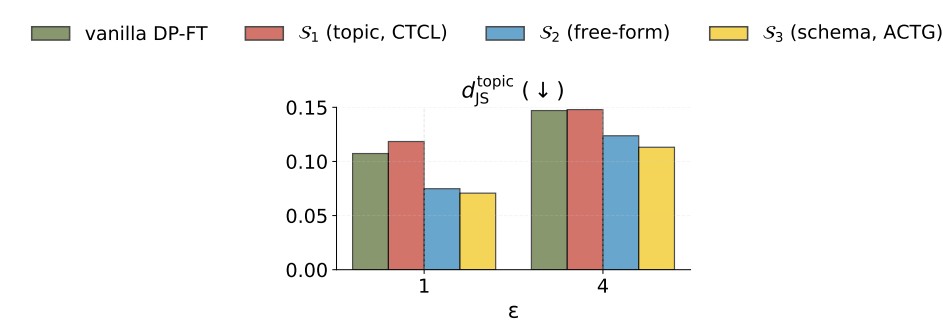

Figure 16: ACTG achieves the best topic distribution matching on bioRxiv.

### F.6 USING IT MODEL FOR CONDITIONAL GENERATION

We additionally experimented with using the instruction-tuned (IT) model `gemma-3-1b-it` as the base model for performing DP-FT to train $G_{x|f}$, instead of the pretrained (PT) model `gemma-3-1b-pt` which we have been using throughout the main paper. The intuition is that an IT model may already have stronger instruction-following ability, and thus can potentially achieve higher IFAcc even under DP. The below results are obtained at $\varepsilon = 1$ on bioRxiv.

**Overall comparison.** Table 2 provides a quantitative comparison across multiple metrics. We found that the IT model performs poorer than the PT model across all metrics. We believe this stems from the following factors:

- *Objective mismatch.* IT models are tuned for instruction following rather than next-token modeling (as performed in DP-FT). Starting from an IT checkpoint gives higher perplexity on the target corpus and less headroom to improve under DP noise. We validate this in Fig. 17, showing that the IT model starts with a higher loss and plateaus at a much worse value than PT.

- *Generic helpfulness vs. domain alignment.* IT tuning bakes in a generic "helpful" style on public data. While this improves general instruction following on simple day-to-day tasks, it does not translate into better alignment with input features in our setting, which requires *domain-specific knowledge*.

Table 2: Comparison of `gemma-3-1b-pt` and `gemma-3-1b-it` as base models for performing DP-FT for conditional text generation.

|  | `gemma-3-1b-pt` | `gemma-3-1b-it` |
| --- | --- | --- |
| **MAUVE** | **0.775** | 0.419 |
| **IFAcc** | **0.534** | 0.503 |
| $d_{\mathrm{JS}}^{f}$ | **0.087** | 0.111 |
| **Classification F1** | **0.726** | 0.716 |

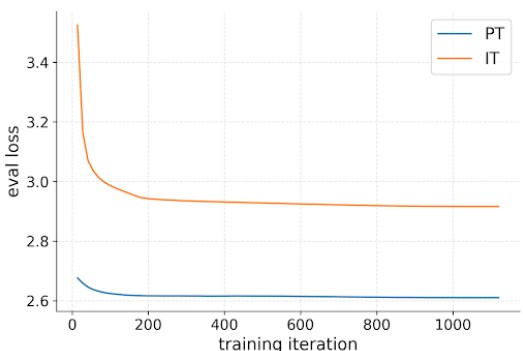

Figure 17: Evaluation loss during DP-FT on `gemma-3-1b-pt` and `gemma-3-1b-it`.

### F.7 IMPACT OF SCHEMA RICHNESS

**A simpler schema.** For the bioRxiv dataset, we consider an alternative simpler schema $\mathcal{S}'_3$. Unlike the $\mathcal{S}_3$ schema (Fig. 7) which consists of *eight* fields that need to be extracted by $M_{\text{oracle}}$, the simpler $\mathcal{S}'_3$ schema contains only *three* ground-truth fields that are directly available or derivable from the dataset: `title`, `category` (the original data columns), and `token_count` (computed from the abstract). We present this schema in Fig. 18.

```
{
    "title": "String", // Title of the paper.
    "category": "< bioengineering | cell biology | bioinformatics | synthetic biology | ecology |
immunology | plant biology | cancer biology | developmental biology | microbiology | biophysics
| genomics | biochemistry | evolutionary biology | pharmacology and toxicology | molecular
biology | scientific communication and education | neuroscience | genetics | systems biology |
physiology | zoology | animal behavior and cognition | pathology | paleontology >", // Category of
the paper.
    "token_count": "Integer", // Number of tokens in the abstract.
}
```

Figure 18: A simpler 3-field schema $\mathcal{S}'_3$ for the bioRxiv dataset

**Instantiation of $G_f$ and $G_{x|f}$.** Since $\mathcal{S}'_3$ contains textual features (e.g., `title`) that AIM cannot process, we adopt DP-FT as the feature generator $G_f$. For the conditional generator $G_{x|f}$, we employ the same DP-FT training procedure used for $\mathcal{S}_3$, learning on the paired (feature, text) set.

**Results.** Fig. 19 presents the end-to-end results on bioRxiv, where we add the new $\mathcal{S}'_3$ (simple schema), as well as $\mathcal{S}_3$ used with DP-FT for feature generation for a fair comparison.

First, we compare $\mathcal{S}'_3$ to the baselines. It significantly outperforms both vanilla DP-FT and CTCL. It also shows an advantage over the $\mathcal{S}_2$ (free-form) approach, indicating that even this simple schema provides a more effective conditioning signal than unstructured summaries.

Next, we directly assess the impact of schema richness by comparing $\mathcal{S}'_3$ (simple schema) with the $\mathcal{S}_3$ (DP-FT+DP-FT) variant. The richer, 8-field $\mathcal{S}_3$ schema outperforms $\mathcal{S}'_3$ in both MAUVE and feature distribution matching, confirming that a more comprehensive feature set is beneficial.

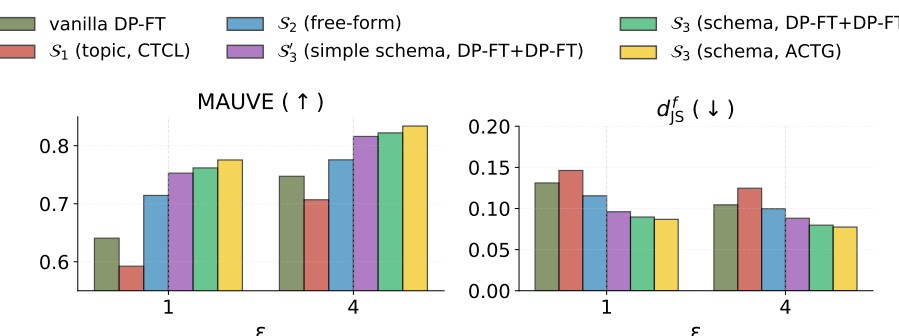

Figure 19: **Impact of schema richness.** We compare the simple 3-field schema ($\mathcal{S}_3'$) against baselines (DP-FT, CTCL), the free-form feature ($\mathcal{S}_2$), the rich 8-field $\mathcal{S}_3$ schema with a DP-FT feature generator, and the full ACTG ($\mathcal{S}_3$ with AIM). Results show that a richer schema outperforms a simple one, while the simpler one already offers clear advantages over the baselines.

### F.8  ROBUSTNESS TO ORACLE CHOICE

**Setup.** To assess whether ACTG depends on a specific proprietary oracle, we replace `Gemini-2.5-flash-lite` with a locally deployed open-source model, `Qwen2.5-32B-Instruct`[24], using the same schema schema $\mathcal{S}_3$ (Fig. 7) with the same prompt template (Fig. 9) for feature extraction.

**Feature-level comparison.** We compute the agreement rate, defined as the ratio of matched fields per sample averaged across the dataset. The extracted features from Qwen and Gemini achieve an agreement rate of 0.69. As a reference, two independent Gemini runs achieve an agreement of 0.84. Because each field contains on average ∼13 categorical options and feature annotation could be inherently ambiguous, we do not expect perfect agreement. More importantly, an agreement of 0.69 already indicates that Qwen captures the core underlying semantics of these fields. This suggests that the extracted features are sufficiently aligned for our use of conditioning—we confirm this with our end-to-end results below.

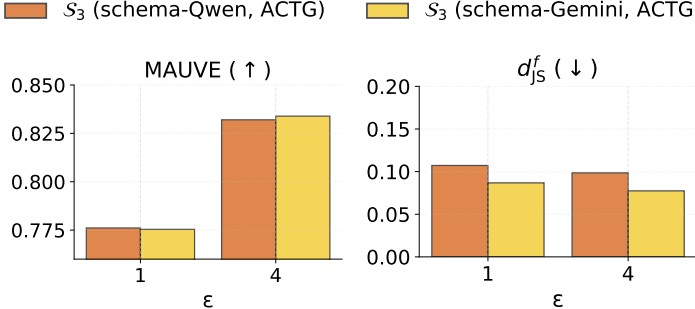

Figure 20: **End-to-end comparison of ACTG using features extracted by** `Qwen2.5-32B-Instruct` **and** `Gemini-2.5-flash-lite`. Dataset: bioRxiv. The Qwen-based pipeline achieves synthetic text quality that closely matches the Gemini-based pipeline, demonstrating robustness to the choice of feature extractor.

**End-to-end results.** We further run the full ACTG pipeline on Qwen-extracted features and present the comparison with the results on Gemini-based pipeline in Fig. 20. The synthetic data produced by the Qwen-based pipeline attains nearly identical MAUVE scores and only minor degradation in feature-distribution matching (due to discrepancy in extracted feature distribution in training data).

---

[24]https://huggingface.co/Qwen/Qwen2.5-32B-Instruct

These results show that ACTG is robust to the choice of feature extractor and can be effectively used with reasonably capable open-source models.

### F.9 AUG-PE WITH A MORE POWERFUL PROPRIETARY MODEL

To further analyze the Aug-PE baseline, we perform an additional experiment with another powerful, state-of-the-art model, **Gemini-2.5-flash-lite**. Fig. 21 plots the performance of the model, compared with Qwen2.5-7B-Instruct across 10 PE iterations.

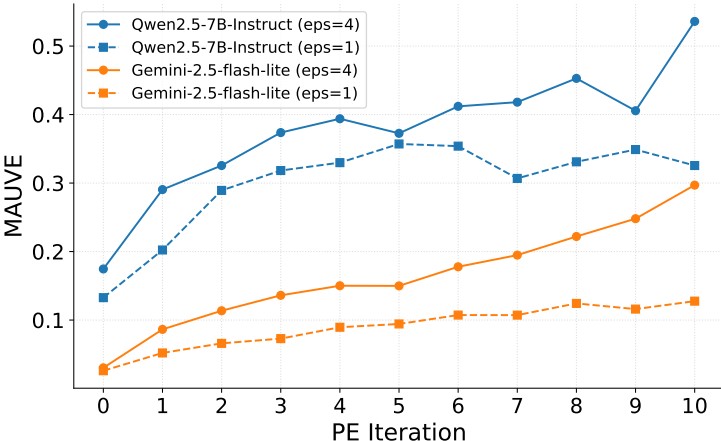

Figure 21: **Performance of Aug-PE using two different models: Qwen2.5-7B-Instruct and Gemini-2.5-flash-lite.** Dataset: bioRxiv. The gap highlights that PE's effectiveness critically depends on the alignment of the model's initial population with the target domain, not just its general capability.

The results are striking: Aug-PE with Qwen2.5-7B-Instruct consistently and significantly outperforms Aug-PE with Gemini-2.5-flash-lite at both privacy levels. This finding strongly indicates that: the performance of PE is less dependent on the model's raw general-purpose capability and far more dependent on how well its initial "out-of-the-box" generations align with the private target distribution. As the figure shows, Qwen's initial population (iteration 0) is substantially better aligned with the private data, providing a strong starting point. Gemini, despite its capabilities, starts with a poorer alignment, and the PE process fails to close this significant gap. This confirms that simply using a "more powerful" model does not guarantee better PE performance; initial domain alignment is the critical factor.

### F.10 EXPERIMENTS ON A LARGER MODEL GEMMA-3-4B-PT

In the main paper, we conducted systematic ablations using a single model, gemma-3-1b-pt. Here, we extend the evaluation to a larger model, **gemma-3-4b-pt**[25] , and show that the performance gains of ACTG persist at this larger scale.

Fig. 22 compares ACTG with the baselines (DP-FT and CTCL). The larger model improves the absolute performance of all methods, giving higher MAUVE scores and better distributional alignment compared to Fig. 2. ACTG nevertheless maintains a clear and substantial advantage over the baselines.

---

[25]https://huggingface.co/google/gemma-3-4b-pt

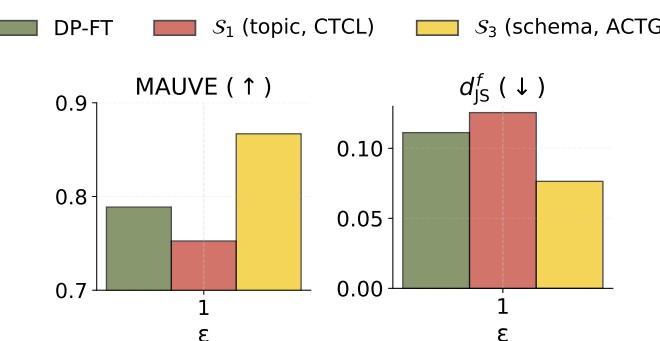

Figure 22: **Comparison of our ACTG with baselines (DP-FT, CTCL) on a larger model** `gemma-3-4b-pt` on bioRxiv, demonstrating its persistent performance advantage at scale.

### F.11    SIGNIFICANCE OF RESULTS

We perform *three* independent runs with different random seeds for the downstream evaluation (classification F1) on bioRxiv. We report the mean and standard deviation for these runs in Fig. 23.

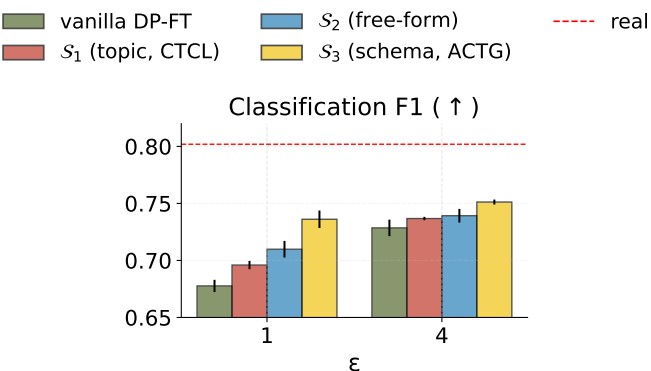

Figure 23: **Mean and standard deviation (black error bars) for the downstream evaluation** (classification F1 for bioRxiv) over *three* independent runs. The small variance and clear separation between ACTG and the baselines demonstrate the statistical significance of our gains.

As the results in Fig. 23 show, the variance across runs is low for all methods. More importantly, the performance gap between ACTG and the baselines is substantially larger than the standard deviation. This analysis confirms that our reported gains on downstream tasks are robust and statistically significant.

### F.12    EXAMPLE OF REWARD HACKING

Fig. 24 illustrates why the generated text in Fig. 4(c) receives a perfect score of 8/8 from $M_{\text{oracle}}$. Although the output is a very short TL;DR-style sentence, it explicitly satisfies every input field: the abstract mentions the correct research area, organism, data type, and focus scale, while also matching the expected approach, sample size, and research goal. Because each criterion is checked independently, the text achieves full credit despite lacking the length, detail, and stylistic fidelity of a proper scientific abstract. This example highlights how RL training can exploit the rubric reward, producing degenerate outputs that maximize score without preserving textual quality.

### F.13    ANALYSIS ON BEST-OF-$N$ DATA

Fig. 25-(left) shows the maximum score difference across candidates for each prompt. Most prompts exhibit large variation, confirming that best-of-$N$ has substantial room to improve over random

- **primary_research_area**: "Neuroscience" ✓ → The abstract mentions **"synaptic plasticity"** and **"spatial memory formation,"** which are core topics in the field of neuroscience.

- **model_organism**: "Drosophila melanogaster" ✓ → The abstract explicitly names the model organism, **"Drosophila."**

- **experimental_approach**: "Wet Lab Experimentation" ✓ → The phrase **"We experimentally evaluated"** directly reflects a hands-on, experimental approach consistent with wet lab work.

- **dominant_data_type**: "Phenotypic / Behavioral" ✓ → **"Spatial memory formation"** is a behavioral or phenotypic trait that is observed and measured in an organism.

- **research_focus_scale**: "Cellular" ✓ → **"Synaptic plasticity"** refers to the ability of synapses (the junctions between nerve cells) to strengthen or weaken over time, which is a phenomenon studied at the cellular level.

- **disease_mention**: "No Specific Disease Mentioned" ✓ → The abstract focuses on fundamental biological processes and does not mention any specific disease.

- **sample_size**: "Relies on Cell/Animal Replicates" ✓ → The study of **"Drosophila"** confirms the use of an animal model, which inherently relies on replicates for experimental validity.

- **research_goal**: "Investigating a mechanism" ✓ → The sentence structure, **"evaluated whether** [process A] **is preserved by modulating** [process B]," describes an investigation into the relationship between two processes, which is a form of investigating a mechanism.

Figure 24: Detailed breakdown of why the TL;DR-style generation in Fig. 4(c) receives a perfect score.

sampling by consistently selecting the strongest candidate. Fig. 25-(right) reports per-rank IFAcc. The highest-ranked candidate (rank 1) achieves an average accuracy above 0.7, which is significantly higher than random samples. Together, these results validate best-of-$N$ sampling as an effective way to distill a cleaner and higher-quality dataset without additional privacy cost.

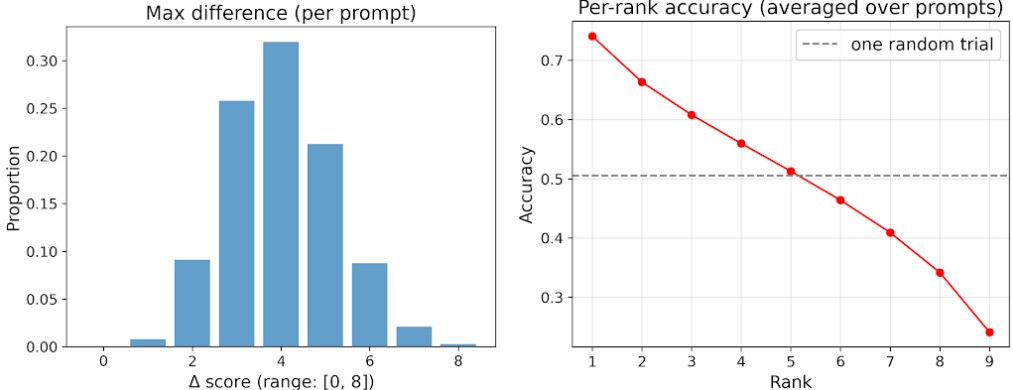

Figure 25: Analysis of best-of-$N$ sampling. **(Left)** Distribution of max score difference per prompt, showing substantial room that best-of-$N$ can exploit. **(Right)** Per-rank IFAcc, demonstrating that higher-ranked candidates can be significantly better than random samples.

## F.14 ADDITIONAL ABLATIONS ON ANCHORED RL

We further ablate the Anchored RL approach to disentangle the benefits of different design choices. In addition to ACTG-(A)RL, we introduce **ACTG-SFT**, where the conditional generator is fine-tuned directly on the anchor dataset ($D_{SFT_1}$ or $DSFT_N$) without reinforcement learning. This setup allows us to isolate the contribution of RL versus the quality of the anchor data itself.

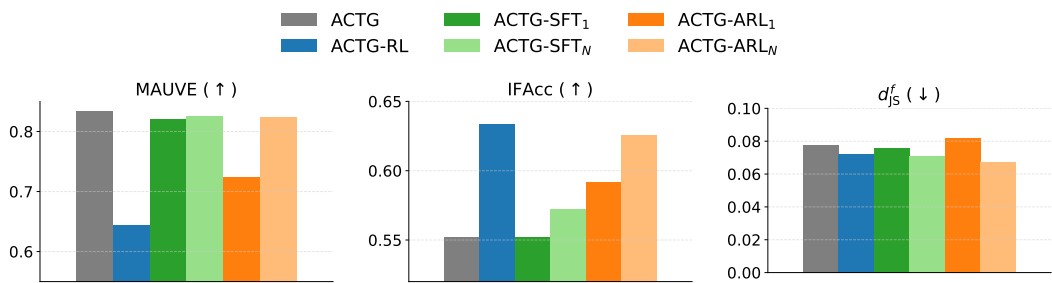

Figure 26: Ablation studies on Anchored RL, where we vary training data and training approaches.

**RL vs. SFT.** Fig. 26 shows that the ARL variants (orange) consistently outperform their SFT counterparts (green) in terms of IFAcc. This highlights the added value of reinforcement learning on anchored data: beyond what supervised fine-tuning alone can capture, RL further boosts fine-grained control.

**Best-of-$N$ sampling.** Comparing the dark versus light hues, we see that models trained on $D_{SFT_N}$ substantially outperform those trained on $D_{SFT_1}$. This confirms the importance of best-of-$N$ sampling: using higher-quality anchors provides a much stronger training signal.

## G COMPUTE RESOURCES AND RUNTIME

**Compute Resources.** All experiments were conducted on one of the following two multi-GPU node configurations:

- **H100 Node:** An 8-GPU node equipped with NVIDIA H100 (80GB) GPUs. This configuration was utilized for experiments involving DP-FT.
- **A100 Node:** An 8-GPU node equipped with NVIDIA A100 (40GB) GPUs. This configuration was used for all other experiments, including Aug-PE, AIM, and variants of RL.

**Runtime.** The approximate runtime for the main stages of our experimental pipeline are as follows:

- **[Inference] Feature extraction**: Querying $M_{oracle}$ (`gemini-2.5-flash-lite`) to extract features according to $\mathcal{S}_3$ for a dataset of 5k samples: $\sim$ 3 minutes.
- **AIM learning and generation**: 5–30 minutes on a single A100 GPU, depending on the concrete $\varepsilon$ and `pgm_iters` used.
- **DP-FT for conditional generation**: $\sim$ 4 hours on the 8-GPU H100 node.
- **Anchored RL training**: $\sim$ 24 hours on the 8-GPU A100 (40GB) node ($\approx$ 6 hours on an 8-GPU H100 (80GB)).
- **[Inference] (Conditional) text generation via sampling from an LLM**: Generating $n = 5$k samples from a fine-tuned `gemma-3-1b` model (context length 512): $\sim$ 20 minutes on a single A100 GPU.
- **[Inference] Best-of-$N$ sampling**: We generate an anchor dataset of size $|D_{SFT}| = 50$k with $N = 9$. The total runtime is $\sim$3.5 hours on an 8-GPU A100 node. We then score the 50k$\times$9 generations by querying $M_{oracle}$; the runtime is about 3 hours. This cost is incurred only once prior to ARL training.

