# OpenReview forum: "Differentially Private Conditional Text Generation with RL-Boosted Control"
_ICLR.cc/2026/Conference — Submitted to ICLR 2026_

### Official Review · Reviewer_g64a · 2025-10-27

**Soundness:** 2
**Presentation:** 3
**Contribution:** 2
**Rating:** 4
**Confidence:** 4

**Summary:**

This paper introduced a hierarchical framework (with ACTG as the optimal configuration) and a novel Anchored RL recipe that, together, form our end-to-end algorithm ACTG-ARL.
Experimental results show improvements over strong baselines (DP-FT, CTCL, Aug-PE) across two datasets and three privacy budgets. The paper is well-motivated, with ablation studies that validate each design choice. It contributes to advancing DP text generation by emphasizing fine-grained controllability—a valuable new dimension alongside utility and privacy.

**Strengths:**

The paper presents a novel and well-structured framework (ACTG) for differentially private (DP) text generation, which decomposes the synthesis process into feature learning and conditional text generation. This modular approach leads to improved privacy-utility trade-offs and provides interpretability benefits.

The introduction of Anchored Reinforcement Learning (ARL) further enhances instruction-following capabilities under privacy constraints, successfully addressing the problem of reward hacking that often plagues RL-based alignment.

**Weaknesses:**

1.	This paper aims to design a differentially private text generator, however, the argument for differential privacy is not sufficient:

①	Authors say Stage 0 doesn’t consume any privacy budget. Why treat Stage 0 with a trusted component and other stages not. I didn’t find the discussions in Appendix C.1

②	How to employ the framework and how to prove the framework is differentially private are not clear. For the DP feature generator, DP-FT is a text generator, how to generate feature here? For DP conditional generator, how to perform DP-FT, and the method( “prompting a powerful LLM…”) didn’t protect privacy.

③	The usage of ARL is not clear. Authors did not the illustration of privacy for this step.

2.	The baseline coverage could be broader. The paper mainly compares against CTCL, DP-FT, and Aug-PE; including recent diffusion-based or graphical-model-based DP synthesizers (e.g., Ochs & Habernal 2025; DeSalvo et al. 2024) would further solidify the empirical claims.

3.	Some technical details—such as PPO background, reward signal stability, and computational overhead of best-of-N sampling—could be expanded for clarity. Additionally, all experiments are performed using a single model size (gemma-3-1b-pt), leaving scaling behavior unexplored.

4.	while ARL is effective, its reliance on an LLM oracle for annotation and evaluation may limit reproducibility or accessibility for smaller labs.

5.	The conclusion in Figure 4(a) does not illustrate the issue because the difference in ε=1/∞ is significant. (The experiment for eps=∞ does not illustrate the issues, as in this case, this is no privacy guarentee.)

**Questions:**

1.	How sensitive is the performance of ARL to the hyperparameter γ and the number of candidates N in best-of-N sampling?

2.	Could the authors clarify the computational cost of the ARL fine-tuning stage relative to ACTG and ACTG-RL?

3.	Is it possible to integrate diffusion-based DP generators into the Stage-1 feature generation step to improve diversity?

4.	How would the proposed method scale if larger or smaller base models were used?

5.	In the ablation study of 5.3.3.2, why is it said that ground-truth features of D^x_priv is not available? This is just a comparative experiment.

---

> ### Author Response · Authors · 2025-11-16
> **Rebuttal (1 / 2)**
>
> Thank you for the review. We are glad that the reviewer appreciates the technical contributions of our work. Below we address your comments.
>
> ---
> ### **Argument for DP.**
> We provide a point-to-point response to the reviewer’s questions.
>   - **Feature extraction.** As discussed in Appendix C.1, “In our main experiments, we assume a threat model in which the server-hosted model is trustworthy. This means that sharing data with the server does not lead to a privacy breach, consistent with the policies of major LLM providers.” From an information flow perspective, we assume the pre-processing step from **private data → private feature** is trusted, whereas all subsequent stages that contribute to the **final synthetic data** need to be protected by DP.
>   - **DP guarantees.** DP-FT is essentially DP-Adam in our setup, and the core privacy mechanism is composition of subsampled Gaussian. AIM achieves a $\rho$-zCDP guarantee. We use standard privacy accountants to compute the suitable parameters (e.g., noise multiplier in DP-Adam) at a given privacy level. Details of privacy accounting are provided in Appendix C.2.
>   - **DP-FT for feature generation.** We first construct the feature dataset by (1) converting each feature record into a textual JSON-style string and (2) prepending a common prefix (e.g., "Please generate a structured JSON summary of a scientific abstract, including the following fields: primary_research_area, model_organism, experimental_approach, dominant_data_type, research_focus_scale, disease_mention, sample_size, research_goal."). We then perform **DP fine-tuning** on this feature dataset, where the loss is computed only on the feature tokens. At inference time, the fine-tuned model is prompted with the same prefix to generate synthetic features.
>   - **DP-FT for conditional generation.** This stage is similar to the previous one. We construct the dataset by wrapping each feature (JSON-style string) within a prompt that serves as conditioning for the corresponding text sample. We then apply **DP fine-tuning**, with the loss computed only on the text portion. During generation, the fine-tuned model is prompted with the **synthetic feature** produced in the previous stage to generate the final synthetic text.
>   - **Directly prompting an LLM for conditional generation.** Prompting an LLM with DP synthetic features does not incur *additional* privacy cost, because it does not further touch the private data.
>   - **Anchored RL (ARL).** As clearly stated in Section 4.1 (“Boosting control via RL”), our RL training phase requires **no additional privacy budget**, since both the prompts and reward signals do not rely on access to the private data.
>
> ---
> ### **Baseline coverage.**
>
> The main finding of *Ochs & Habernal (2025)* is that DP fine-tuning a diffusion model performs significantly worse than LLM-based approaches. This justifies why we did not include the diffusion-based approach as a baseline. Nonetheless, we have already cited this work in the related work section. Regarding *SoftSRV (DeSalvo et al., 2024)*, we note that it does not involve DP at all and therefore does not qualify as a relevant baseline. That said, we have already discussed it in the related work section for completeness.
>
> ---
> ### **Technical details.**
>
> The details of PPO are not central to the main focus of this work and are therefore deferred to **Appendix C.3**. The quality of the reward signal corresponds to the annotation error, which we have analyzed in Section 3.3.2 (Ablation studies). In addition, we have reported the computational cost of best-of-N sampling in **Appendix G**. We note that it incurs only a one-time cost.

---

> ### Author Response · Authors · 2025-11-16
> **Rebuttal (2 / 2)**
>
> ---
> ### **Scaling behavior.**
>
> We evaluated ACTG on a larger model, gemma-3-4b-pt, and compared it with CTCL and vanilla DP-FT. The results are in **Appendix F.10**. While the larger model improves the absolute performance of all methods, ACTG maintains a clear and substantial advantage over both baselines. This shows that the performance gains of ACTG persist at a larger scale.
>
> We emphasize that studying scaling behavior across models is an **important but orthogonal** to the current work; this is clearly acknowledged in the paper as a limitation and part of future work. In our paper, we intentionally focus on a fixed model to enable systematic and controlled ablation studies across all components of our hierarchical framework, as well as the subsequent RL training stage. Furthermore, as Reviewer 9s83 also noted, “the codebase is well structured, sufficiently documented, and provides a strong foundation for others to build upon”. The strong reproducibility of our codebase means that evaluating our approach on larger or different models is straightforward; it simply requires additional compute resources.
>
> ---
> ### **Choice of oracle LLMs.**
> To evaluate whether our approach is tied to the capabilities of oracle LLMs, we add a new experiment using a locally deployed open-source model, Qwen2.5-32B-Instruct, as the feature extractor. The setup and full results are provided in Appendix F.8. In short, running the ACTG pipeline with Qwen-extracted features yields **almost identical** MAUVE scores to those obtained using Gemini-extracted features (0.8320 vs. 0.8339 at $\varepsilon = 4$). This demonstrates that ACTG is robust to the choice of feature extractor and can be effectively used with reasonably capable open-source models.
>
> ---
> ### **Figure 4(a).**
> The significant difference in $\varepsilon=1, \infty$ exactly shows that when going from non-DP ($\varepsilon = \infty$) to DP ($\varepsilon=1$), the instruction following ability drops a lot.
>
> ---
> ### **Sensitivity of hyperparameters.**
> We have identified hyperparameters that perform well across different setups: for $\gamma$, a linear decay schedule is recommended with detailed hyperparameters in Appendix E.4; using a start value in the range of 0.5–2 and an end value in the range of 0.1–0.5 generally works. For $N$, we found that $N=9$ strikes a good balance between efficiency and performance.
>
> ---
> ### **Computational cost of ARL.**
> We provide the detailed runtime of different stages in Appendix G. The cost of the ARL stage is about 1.5–2x of the ACTG stage. The overhead of ARL relative to standard RL is small (less than 10%), corresponding to the cost of calculating $\mathcal{L}_{\text{SFT}}$ in each forward pass.
>
> ---
> ### **Diffusion-based feature generators.**
> The feature generator aims to preserve the true feature distribution in the private dataset, rather than simply improving coverage. A core strength of DP tabular synthesizers is their targeted use of privacy budget: they allocate privacy only to a specified set of attributes, whereas DP-FT spreads privacy budget across all tokens in the sequence (whether using LLMs or diffusion models). As a result, specialized DP tabular synthesizers are generally better suited for feature generation.
>
> ---
> ### **Ground-truth features.**
> For example, in the bioRxiv dataset, even experts in related fields may find it challenging to accurately determine a paper abstract’s primary research area among the 20 categories defined in the schema. Moreover, obtaining perfectly consistent expert annotations at scale is practically infeasible.
>
> ---
> ### **Final words.**
> We hope the above clarifies your concerns. We would be happy to discuss further if there are remaining questions. If our responses address your points, we would appreciate your reconsideration of the rating in support of our work. Thanks again for your time.

---

> > ### Author Response · Authors · 2025-11-24
> > **Would you mind taking a look at our rebuttal and global response?**
> >
> > Dear Reviewer g64a,
> >
> > We hope you are doing well. It’s been over a week since we have posted the rebuttal. We have run additional experiments and provided detailed clarifications that we believe meaningfully address the concerns raised in the initial reviews. We have also summarized the main contributions of our work in the global response. We believe the submission offers an important contribution to the field of DP synthetic text generation and will be of broad interest to the community.
> >
> > We understand this is a busy time of year, but we would be grateful if you could take a look at our rebuttal and global response at your earliest convenience.
> >
> > Thank you very much for your time.
> >
> > Authors of submission 6311

---

> ### Comment · Reviewer_g64a · 2025-11-26
>
> Thank authors for your time and effort in the response. I appreciate your feedback but maintain my score, primarily considering the paper's core contribution.

---

> > ### Author Response · Authors · 2025-11-26
> > **Request for clarification**
> >
> > Thank you for engaging with our response. May we ask two clarifying questions?
> >
> > - Do you feel that we have sufficiently addressed the concerns you originally raised?
> >
> > - Are there specific points in the global response where you disagree with our reasoning or feel that we are overstating our claims?
> >
> > Our intention in the rebuttal is to enable a constructive discussion about the paper’s contributions. We are very open to further discussion, but it would be most helpful if you could point to the specific aspects where our views diverge, beyond the high-level comment regarding the “core contribution”.

---

> > > ### Author Response · Authors · 2025-11-27
> > > **Friendly reminder on ICLR rebuttal guidelines**
> > >
> > > We would like to follow up once more in light of ICLR’s guidance for the rebuttal phase.
> > >
> > > As a reminder, the conference instructions ask reviewers to
> > > (a) acknowledge the rebuttal, and
> > > (b) **"precisely identify which of your concerns have been addressed, which have not been addressed and why. Please also update your review score if your concerns have been addressed.”**
> > >
> > > Your recent reply acknowledges our response, but it does not engage with part (b). The statement “maintain my score, primarily considering the paper’s core contribution” does not specify which concerns you feel remain unresolved, nor why our clarifications or additional results may not have been sufficient.
> > >
> > > We also want to emphasize that we do not view the rating alone as the main focus of the rebuttal process. **Our goal in engaging actively is not to pressure for a score increase, and we fully respect it if reviewers maintain their scores after substantive discussion. What matters far more to us is having a constructive technical exchange that helps us understand any remaining disagreements and improve the work accordingly.**
> > >
> > > This message is intended for all reviewers of our submission. Thank you for taking the time to read it.

---

### Official Review · Reviewer_9s83 · 2025-10-31

**Soundness:** 4
**Presentation:** 4
**Contribution:** 3
**Rating:** 6
**Confidence:** 4

**Summary:**

The paper presents a new hierarchical framework for generating high-quality synthetic text under differential privacy (DP). Prior approaches to DP synthetic text generation often struggle with preserving key statistical attributes, suffer from significant utility loss due to injected noise, and lack fine-grained control during generation.To address these issues, the authors decompose the DP synthetic text generation task into two subtasks: feature learning and conditional text generation. Their framework uses a rich tabular schema as a feature representation, a DP tabular synthesizer to ensure privacy during feature learning, and a DP fine-tuned conditional generator for text synthesis. Another key contribution is Anchored Reinforcement Learning (Anchored RL), a post-training method that enhances the instruction-following capability of the conditional generator, ACTG, under DP constraints. Empirically, the proposed method improves both text quality and control compared to prior work. It also offers strong privacy guarantees, allowing the resulting DP synthetic datasets to be reused without additional privacy costs.

**Strengths:**

- The paper is clearly written and easy to follow, with a logical flow that makes the main ideas and contributions understandable.

- The arguments are well structured, and the overall organisation effectively supports the proposed framework and experimental results.

- The inclusion of code significantly enhances the validity and reproducibility of the work. The codebase is well structured, sufficiently documented, and provides a strong foundation for others to build upon.

**Weaknesses:**

- The reported performance improvements, while consistent, are modest compared to baselines such as vanilla DP-FT and CTCL. The authors could strengthen their claims by including statistical measures of variability (e.g., variance bars or standard deviations) in the plots to show robustness across multiple runs.

- It would be valuable to include an additional baseline that applies Anchored RL directly to vanilla DP-FT. This would help isolate and clarify the contribution of Anchored RL to the overall performance gains.

- All experiments were conducted using a single model, which the authors acknowledge as a limitation.


The paper presents a solid contribution with clear writing, strong methodological grounding, and commendable reproducibility. However, I remain uncertain about how much of the reported improvements can be attributed to noise, given the absence of variance bars or discussion of experimental variability. Therefore, I am assigning a score of 6. I would be open to reconsidering my score based on the rebuttal, particularly if the authors can clarify the robustness of their results across multiple runs and provide additional statistical evidence supporting the observed gains.

**Questions:**

- I could not locate the script used to produce Figure 6, Appendix C (schema identification) in the provided codebase. Including this script would make the work fully reproducible and ensure that others can replicate the S3 approach end-to-end.

- Could the authors clarify whether the number or quality of extracted features influences the effectiveness of the S3 approach? An ablation study examining how varying the number of features impacts performance would provide deeper insights into the method’s sensitivity and limitations.

---

> ### Author Response · Authors · 2025-11-16
> **Rebuttal**
>
> Thank you for the review. We greatly appreciate that you took the time to examine our code and recognized it as “well structured, sufficiently documented, and providing a strong foundation for others to build upon”. We view this as a strong endorsement of our work. Below we address your comments.
>
> ---
> ### **Significance of performance improvement.**
>
> We agree that including variance bars would help assess the statistical significance of our improvements. There are two sources of randomness: (1) the randomness in generating the synthetic dataset, and (2) the randomness in evaluating the synthetic dataset. The first source is **usually not accounted for in prior work**, as the ultimate goal is to produce high-quality synthetic data rather than to characterize variance across generations. The second source arises **only in downstream evaluations**, particularly in training machine learning models, whereas the fidelity metrics we report (MAUVE and feature distribution matching) are deterministic.
>   - For downstream evaluation metrics (classification F1), we perform *three* runs with different random seeds and report the mean and standard deviation in updated **Appendix F.11**. The result shows that our reported gains on downstream tasks are robust and statistically significant.
>   - Regarding the MAUVE score, we note that **ACTG surpasses prior state-of-the-art methods by a large margin**. For example, ACTG achieves a relative improvement of 20% over the best among {CTCL, vanilla DP-FT, Aug-PE} (78% vs. 64%) at $\varepsilon=1$ on bioRxiv, representing a clear and significant improvement. Moreover, our additional experiments with a larger model gemma-3-4b-pt (**Appendix F.10**) and using Qwen-extracted features (**Appendix F.8**) both show the consistent edge of our ACTG over the baselines.
>
> ---
> ### **Combining Anchored RL and vanilla DP-FT.**
>
> As noted at the beginning of Sec. 4, using RL to enhance control is enabled by the conditional generation framework and the design of ACTG, where tabular features serve as explicit and verifiable rewards. These features are **not** available in vanilla DP-FT, making it incompatible with RL. In the early stage of this project, we tried applying RL directly by privately aligning generated text with private text, but this approach suffered from severe reward hacking and mode collapse.
>
> ---
> ### **Using a single model for evaluation.**
>
> We evaluated ACTG on a larger model, gemma-3-4b-pt, and compared it with CTCL and vanilla DP-FT. The results are in **Appendix F.10**. While the larger model improves the absolute performance of all methods, ACTG maintains a clear and substantial advantage over both baselines. This shows that the performance gains of ACTG persist at a larger scale.
>
> We emphasize that studying the scaling behavior across models is an **important but orthogonal direction** to the current work; this is clearly acknowledged in the paper as a limitation and part of future work. In our paper, we intentionally focus on a fixed model to enable systematic and controlled ablation studies across all components of our hierarchical framework, as well as the subsequent RL training stage. Furthermore, the strong reproducibility of our codebase means that evaluating our approach on larger or different models is straightforward; it simply requires additional compute resources.
>
> ---
> ### **Script for Figure 6.**
> Thank you for noting this! We have updated our GitHub repository to include the relevant code under ``annotation/``.
>
> ---
> ### **Richness of features.**
> Earlier in this project, we examined how the richness of features affects the quality of DP synthetic data. For bioRxiv, we compared two feature designs: the full tabular schema (8 attributes) and a simpler design consisting of 3 native attributes from the dataset (title, category, and token count). We applied DP fine-tuning to learn feature generators for both designs to ensure a fair comparison. The results (**Appendix F.7**) show that the full tabular schema yields stronger performance than the simpler one, and both outperform vanilla DP-FT. This supports the view that a richer schema design can more faithfully capture the key information present in the original private text.
>
> ---
> ### **Final words.**
> Once again, we sincerely thank you for the thoughtful review and for taking the time to examine our codebase. We hope the above clarifies your concerns, and we would be glad to discuss further if any questions remain. We would greatly appreciate your support for our work.

---

> > ### Author Response · Authors · 2025-11-24
> > **Would you mind taking a look at our rebuttal and global response?**
> >
> > Dear Reviewer 9s83,
> >
> > We hope you are doing well. It’s been over a week since we have posted the rebuttal. We have run additional experiments and provided detailed clarifications that we believe meaningfully address the concerns raised in the initial reviews. We have also summarized the main contributions of our work in the global response. We believe the submission offers an important contribution to the field of DP synthetic text generation and will be of broad interest to the community.
> >
> > We understand this is a busy time of year, but we would be grateful if you could take a look at our rebuttal and global response at your earliest convenience.
> >
> > Thank you very much for your time.
> >
> > Authors of submission 6311

---

### Official Review · Reviewer_hwX2 · 2025-10-31

**Soundness:** 3
**Presentation:** 2
**Contribution:** 2
**Rating:** 4
**Confidence:** 4

**Summary:**

This paper presents a new approach to generating DP synthetic text using a hierarchical framework. The model is designed to address the challenges of privacy, data utility, and controlled generation. The framework decomposes the generation task into two stages: feature learning and conditional text generation. The authors introduce ACTG (Attribute-Conditioned Text Generation), a method that optimizes both DP synthetic text quality and fine-grained control over the generation. They also propose a post-training method, Anchored RL (ARL), which improves the instruction-following ability of ACTG by addressing control degradation under DP.

**Strengths:**

1.The hierarchical framework (ACTG) combines a DP tabular synthesizer with a DP fine-tuned conditional generator. This separation of tasks allows for improved optimization.

2.The introduction of ARL to address instruction-following is a significant contribution. It demonstrates the balance between privacy and control in DP settings, achieving better performance than previous methods.

3.ACTG-ARL outperforms prior methods in multiple metrics, including MAUVE and attribute distribution matching. This provides solid evidence of its practical utility.

**Weaknesses:**

1.The framework depends heavily on accurate feature extraction, but does not provide a detailed description of the feature extractor, especially regarding how attributes are selected or normalized before DP processing.

2.The difference between this work and CTCL isn’t made very clear. Although the paper introduces some refinements, it largely follows CTCL’s existing structure of “public pretraining → private fine-tuning → synthetic generation.”  The overall idea and workflow feel quite similar, and the updates seem more like incremental technical improvements than a fundamentally new approach.

3.The evaluation is limited to one model configuration, leaving open the question of how well the approach generalizes across different model sizes and capacities.

**Questions:**

1.The evaluation is conducted on a single model configuration, but the chosen model and experimental setup raise questions. In the Aug-PE stage, the paper replaces the original GPT-3.5 (used in prior work) with Qwen, which undermines one of the core advantages of the approach,its compatibility with black-box LLMs. This substitution also makes it unclear whether the reported improvements are due to the framework itself or to differences in the underlying model. Furthermore, Aug-PE should theoretically outperform DP-FT given its reinforcement of privacy-preserving generalization, yet this expected advantage is not reflected in the experimental results.
2.How can the trade-off between control and text fidelity be better balanced during the joint optimization of SFT and ARL?

---

> ### Author Response · Authors · 2025-11-16
> **Rebuttal (1 / 2)**
>
> Thank you for the review. We are glad that the reviewer appreciates the technical contributions of our work, particularly Anchored RL. Below we address your comments.
>
> ---
> ### **Details of feature extraction.**
>
> Due to space constraints, the full details are provided in Appendix C.1. Briefly, the process involves:
>   - Prompting an LLM to design a tabular schema consisting of multiple attributes, each with a set of categorical options;
>   - Prompting the oracle LLM to annotate each private text sample according to the schema by selecting the best matching option for each attribute.
>
> The prompts for schema design and extraction are shown in Figures 6 and 9, and the resulting schemas for the bioRxiv and PMC-patients datasets are provided in Figures 7 and 8.
>
> As discussed in our response to Reviewer RyGE, the schema-design prompt can be easily adapted to any custom dataset. We have also analyzed extraction error in the “Ablation Studies” section (Sec. 3.3.2) and examined the use of a locally deployable, open-source model for extraction following Reviewer RyGE’s suggestion (**Appendix F.8**), showing that it barely affects the performance of ACTG.
>
> ---
> ### **Comparison with CTCL.**
>
> We apologize for the confusion. We clarify how our work differs from CTCL at the beginning of Sec. 3.1. The limitations of CTCL motivate us to generalize it into a broader and more flexible hierarchical framework and to systematically analyze its design choices. Our algorithm, ACTG, differs from CTCL in several fundamental ways:
>
>   - Instead of “public pretraining → private fine-tuning → synthetic generation”, it follows the “feature design & extraction → feature learning → conditional generation” pipeline.
>
>   - It introduces insights from **DP structured data synthesis** into unstructured synthesis, leveraging the strengths of both and demonstrating the power of this hybrid approach.
>
>   - It achieves **consistent and significant performance improvements** over the state of the art, whereas CTCL even underperforms vanilla DP-FT in some settings.
>
> Additionally, as you noted, “the introduction of ARL to address instruction-following is a significant contribution”. Elevating *control* as a third key dimension alongside utility and privacy in DP synthetic text generation is an important conceptual advancement, and the ARL algorithm is a major technical contribution.
>
> Taken together, we believe our work represents a substantial conceptual and technical advance over prior work, offering new perspectives and methods for DP synthetic text generation. We therefore respectfully disagree with the characterization of our contribution as “incremental technical improvements”.
>
> ---
> ### **Using a single model for evaluation.**
>
> We evaluate ACTG on a larger model, gemma-3-4b-pt, and compared it with CTCL and vanilla DP-FT. The results are in **Appendix F.10**. While the larger model improves the absolute performance of all methods, ACTG maintains a clear and substantial advantage over both baselines. This shows that the performance gains of ACTG persist at a larger scale.
>
> We emphasize that studying the scaling behavior across models is an **important but orthogonal direction** to the current work; this is clearly acknowledged in the paper as a limitation and part of future work. In our paper, we intentionally focus on a fixed model to enable systematic and controlled ablation studies across all components of our hierarchical framework, as well as the subsequent RL training stage. Furthermore, as Reviewer 9s83 also noted, “the codebase is well structured, sufficiently documented, and provides a strong foundation for others to build upon”. The strong reproducibility of our codebase means that evaluating our approach on larger or different models is straightforward; it simply requires additional compute resources.

---

> > ### Author Response · Authors · 2025-11-16
> > **Rebuttal (2 / 2)**
> >
> > ---
> > ### **Choice of model in Aug-PE.**
> >
> > We first clarify that *"Aug-PE should theoretically outperform DP-FT given its reinforcement of privacy-preserving generalization"* is a **misconception**. Prior work (CTCL) already showed that DP fine-tuning a language model with O(100M) parameters surpasses Aug-PE by a large margin. Their comparison used Aug-PE’s self-reported results from GPT-3.5. Our findings are consistent with CTCL and further demonstrate that PE performs substantially worse than other fine-tuning based methods.
> >
> > We also note that although GPT-3.5 is larger than Qwen-2.5-7B-Instruct, it is not necessarily stronger. As reported in the GPT-4 technical report (https://arxiv.org/pdf/2303.08774), GPT-3.5 scores 70% on MMLU. In contrast, Qwen-2.5-7B-Instruct reaches 74.2% (https://qwen.ai/blog?id=qwen2.5-llm).
> >
> > Finally, we conducted PE using Gemini-2.5-flash-lite as the base model, which is substantially more capable than GPT-3.5. The full results are in **Appendix F.9**. The findings are striking: **Aug-PE with Qwen-2.5-7B-Instruct consistently and significantly outperforms Aug-PE with Gemini-2.5-flash-lite** at both privacy levels. Examining the MAUVE–iteration curves, we further observe that **Aug-PE’s performance depends far less on the raw capability of the underlying model and instead critically hinges on how well the model’s initial, out-of-the-box generations align with the private data distribution**. Consequently, Aug-PE not only underperforms fine-tuning–based methods by a large margin, but is also substantially more sensitive to the choice of base model.
> >
> > ---
> > ### **Trade-off between control and fidelity.**
> > ARL uses a hybrid RL and SFT objective to balance control and fidelity. We employ a linear decay schedule for the SFT coefficient $\gamma$, starting high to preserve text fidelity and gradually decreasing to allow for steady improvement in instruction following.
> >
> > ---
> > ### **Final words.**
> > We hope the above clarifies your concerns. We would be happy to discuss further if there are remaining questions. If our responses address your points, we would appreciate your reconsideration of the rating in support of our work. Thanks again for your time.

---

> > > ### Author Response · Authors · 2025-11-24
> > > **Would you mind taking a look at our rebuttal and global response?**
> > >
> > > Dear Reviewer hwX2,
> > >
> > > We hope you are doing well. It’s been over a week since we have posted the rebuttal. We have run additional experiments and provided detailed clarifications that we believe meaningfully address the concerns raised in the initial reviews. We have also summarized the main contributions of our work in the global response. We believe the submission offers an important contribution to the field of DP synthetic text generation and will be of broad interest to the community.
> > >
> > > We understand this is a busy time of year, but we would be grateful if you could take a look at our rebuttal and global response at your earliest convenience.
> > >
> > > Thank you very much for your time.
> > >
> > > Authors of submission 6311

---

### Official Review · Reviewer_RyGE · 2025-11-02

**Soundness:** 3
**Presentation:** 3
**Contribution:** 3
**Rating:** 4
**Confidence:** 3

**Summary:**

The authors present ACTG-ARL, a novel framework for differentially private (DP) conditional text generation that aims to improve both the quality of synthetic text and fine-grained control over generation while maintaining strong privacy guarantees. The authors propose a hierarchical framework that decomposes DP synthetic text generation into feature learning and conditional text generation. They also introduce Anchored RL (ARL), a post-training method that uses reinforcement learning (RL) with a supervised fine-tuning (SFT) anchor to boost instruction-following ability and mitigate reward hacking.

**Strengths:**

S1. The authors clearly highlight the weaknesses in existing DP based text generation techniques. The motivation for the paper is strong.

S2. ACTG is modular and hierarchical potentially allowing better privacy utility tradeoffs.

S3. I like the Anchored RL approach for better instruction following in DP-trained conditional generations. The empirical results look strong and the evaluations are well grounded.

**Weaknesses:**

W1. The complexity and computational cost of the overall framework might be challenging to implement. More discussion on this should be added in the paper.

W2. The quality of the generated features and rewards is directly tied to the capabilities of these oracle LLMs. The paper acknowledges this with the discussion on extraction error but doesn't fully explore potential limitations if a less capable or open-source LLM is used as the oracle.

W3. While the structured tabular schema (S3) performs well, the process of designing such a schema (LLM-assisted) is described as "dataset-specific." This raises questions about how much manual effort or expert knowledge is required to create effective schemas for new domains, and whether the LLM assistance is truly robust across highly diverse data.

**Questions:**

See weaknesses

---

> ### Author Response · Authors · 2025-11-16
> **Rebuttal**
>
> Thank you for the review. We are glad that the reviewer found the motivation and technical contributions strong. Below we address your comments.
>
> ---
> ### **Complexity and computational cost.**
>
> Following your suggestion, we have added details on compute resources and runtime in **Appendix G** of the revised draft. The approximate runtime for the main stages of our experimental pipeline are:
>   - **Feature extraction**: 3 minutes on 5k samples
>   - **AIM learning and generation**: 10 minutes on a single A100 GPU
>   - **DP-FT for conditional generation**: 4 hours on an 8-GPU H100 (80GB) node
>   - **[Optional] Anchored RL training**: 24 hours on an 8-GPU A100 (40GB) node (6 hours on an 8-GPU H100 (80GB) node)
>
> The additional cost of ACTG over vanilla DP-FT is small (the first two steps). While Anchored RL training is more expensive, it serves a different purpose (improving control), whereas ACTG alone already achieves high-quality DP synthetic data that significantly outperforms the state of the art.
>
> As Reviewer 9s83 noted, *“the codebase is well structured, sufficiently documented, and provides a strong foundation for others to build upon.”* We hope this clarifies concerns about implementation complexity for future work.
>
> ---
> ### **Choice of oracle LLMs.**
>
> To evaluate whether our approach is tied to the capabilities of oracle LLMs, we add a new experiment using a locally deployed open-source model, Qwen2.5-32B-Instruct, as the feature extractor. The setup and full results are provided in **Appendix F.8**. In short, running the ACTG pipeline with Qwen-extracted features yields **almost identical** MAUVE scores to those obtained using Gemini-extracted features (0.8320 vs. 0.8339 at $\varepsilon = 4$). This demonstrates that ACTG is robust to the choice of feature extractor and can be effectively used with reasonably capable open-source models.
>
> ---
> ### **Manual effort and robustness of schema design.**
>
> We emphasize that our *schema design process* is **largely automatic** and requires minimal manual effort; what we highlighted in the paper is that the *schema itself* needs to be dataset-specific. We have included the details of schema design for a custom dataset in Appendix C.1 (Fig 6) with code in our GitHub repository. In short, the process requires specifying three configuration fields: dataset_description, workload_description, and num_features. The first captures general information about the dataset (e.g., paper abstracts in biological domains); the second describes potential feature usage (can be left empty); empirically, setting num_features=8 offers a good balance between coverage and efficiency.
>
> Using this procedure, we automatically generated schemas for two specialized datasets (biological paper abstracts and clinical notes). Upon manual inspection (Fig. 7-8), we found the schemas well-crafted.
>
> We acknowledge that this procedure may not generalize to arbitrary datasets (e.g., a mixture of bioRxiv and PMC-patients), but the intended use case of DP synthetic data is precisely specialized domains. Moreover, since schema design is a plug-in component of our framework and a minor point compared to the broader insights we aim to convey, we did not heavily optimize it.
>
> ---
> ### **Final words.**
>
> We hope the above clarifies your concerns. We would be happy to discuss further if there are remaining questions. If our responses address your points, we would appreciate your reconsideration of the rating in support of our work. Thanks again for your time.

---

> > ### Author Response · Authors · 2025-11-24
> > **Would you mind taking a look at our rebuttal and global response?**
> >
> > Dear Reviewer RyGE,
> >
> > We hope you are doing well. It’s been over a week since we have posted the rebuttal. We have run additional experiments and provided detailed clarifications that we believe meaningfully address the concerns raised in the initial reviews. We have also summarized the main contributions of our work in the global response. We believe the submission offers an important contribution to the field of DP synthetic text generation and will be of broad interest to the community.
> >
> > We understand this is a busy time of year, but we would be grateful if you could take a look at our rebuttal and global response at your earliest convenience.
> >
> > Thank you very much for your time.
> >
> > Authors of submission 6311

---

### Author Response · Authors · 2025-11-16
**Global response**

We sincerely thank the reviewers for their thoughtful feedback, including comments on the choice of oracle LLM, the use of a single model, and schema design. We have incorporated these suggestions and conducted additional experiments accordingly. All new results are included in the revised draft. Some highlights include:


  - In **Appendix F.8**, we replace Gemini with Qwen2.5-32B-Instruct (an open-source model that can be run locally) for feature extraction, and repeat the ACTG pipeline. The resulting DP synthetic data achieves nearly identical MAUVE scores, demonstrating that ACTG is robust to the choice of feature extractor and can work effectively with reasonably capable open-source models.
  - In **Appendix F.10**, we evaluate a larger model, gemma-3-4b-pt. While scaling up improves performance for all methods, ACTG maintains a clear and substantial advantage over other baselines, confirming that our performance gains persist at larger scales.
  - In **Appendix F.7**, we examine a simpler 3-field schema and find that even this minimal schema offers meaningful conditioning signals and outperforms all baselines. At the same time, a richer schema more faithfully captures key information of the private dataset and therefore performs better.


We are grateful for these suggestions, which help us further strengthen the paper. We view them as complementary to our main contributions and the broader significance of our work, summarized below.


  - **Structured insights in unstructured data synthesis.**
We reframe part of the task of DP text synthesis as DP tabular data synthesis, enabling us to combine the strengths of both fields within a unified hierarchical framework. This perspective of *leveraging structured insights to guide unstructured data synthesis* represents a major conceptual advance.
  - **Substantial and consistent performance improvements.**
Across all datasets, privacy regimes and evaluation metrics, ACTG delivers large, consistent improvements over prior methods. These results validate the effectiveness of our hierarchical design and the hybrid tabular-text synthesis strategy.
  - **Introducing control as a third fundamental dimension.**
Our work is the first to explicitly elevate *control*, alongside utility and privacy, as a third fundamental dimension in DP synthetic text generation. This conceptual shift reframes the problem space and opens new possibilities for fine-grained generation under privacy constraints. Building on this principle, we propose the Anchored RL (ARL) algorithm, which adapts state-of-the-art LLM training and inference techniques to the DP setting, a key technical contribution that operationalizes this new dimension.
  - **Strong reproducibility and community impact.**
As Reviewer 9s83 remarked, our codebase is “well structured, sufficiently documented, and provides a strong foundation for others to build upon”. This reproducibility is intentional: our goal is to make future extensions straightforward. To further support the community, **we have released all of our DP synthetic datasets on our anonymous GitHub repository**.
  - **Critical re-evaluation of prior practices.**
Although not highlighted in the reviews, our paper also revisits prevailing evaluation practices (Remark 3.3) and identifies critical issues, such as reliance on weak embedding models and short context lengths. We address these shortcomings and establish a more rigorous evaluation pipeline for future work.


Taken together, our work introduces new perspectives, tools, and evaluation protocols for DP synthetic text generation. We believe these contributions represent a meaningful step forward and will be of broad interest to the community.

---

### Meta-Review · Area_Chair_2oHQ · 2026-01-06

**Summary:**

Based on the four reviews provided, the concerns are quite consistent across the board, focusing on privacy proofs, statistical rigor, and methodological novelty. Specifically, multiple reviewers highlighted that the experimental scope is too narrow. The novelty against existing work CTCL is also a major concern. In addition, reviewers are also concerned with the practicality of the framework, especially the scalability of creating effective schema in other domains, insufficient baselines and statistical significance of results.

**Reviewer Concerns:**

The rebuttal partially addressed some of the concerns. Yet there are still critical issues outstanding. Three out of four reviewers pointed out the issue with scalability of model size, yet this is not systematically evaluated and not much insights can be drawn, although the quality of the features and rewards in RL in the framework is depending on the capabilities of the oracle LLMs.  These limitations are suggested to be carefully studied and presented. Secondly, the novelty against CTCL, including the details of the proposed feature extractors and the ARL step, and the threat models, need to be presented in detail clearly in the main paper.

**Reviewer Scores:**

Reviewer 9s83: I don't think the reviewer would raise score given that the issues are not fully resolved. The issue with the statistical significance (lack of variance) across the paper's main results are not changed. Only one figure with partial results are provided with statistical STD, which is not comprehensive. The additional baseline results requested by the reviewer are not provided. The problem with using one model for evaluation is not systematically addressed.

Reviewer RyGE：The reviewer may maintain the score given that scalability of model is only narrowly evaluated, and the limitation  for the generalization of the procedure of creating tabular scheme to an arbitrary domain is still an open challenge.

Reviewer hwX2: I find that reviewers' concerns on the novelty against existing approach, the use of single model and lack of technical details are only partially addressed.
Reviewer 9s83: Additional baselines raised by the reviewers are not sufficiently compared.

---

### Decision · Program_Chairs · 2026-01-26

Reject